# LOCAL GRAPH CLUSTERING WITH NOISY LABELS

**Artur Back de Luca, Kimon Fountoulakis, Shenghao Yang**
School of Computer Science, University of Waterloo, Canada
{abackdel,kimon.fountoulakis,shenghao.yang}@uwaterloo.ca

## ABSTRACT

The growing interest in machine learning problems over graphs with additional node information such as texts, images, or labels has popularized methods that require the costly operation of processing the entire graph. Yet, little effort has been made to the development of fast local methods (i.e. without accessing the entire graph) that extract useful information from such data. To that end, we propose a study of local graph clustering using noisy node labels as a proxy for additional node information. In this setting, nodes receive initial binary labels based on cluster affiliation: 1 if they belong to the target cluster and 0 otherwise. Subsequently, a fraction of these labels is flipped. We investigate the benefits of incorporating noisy labels for local graph clustering. By constructing a weighted graph with such labels, we study the performance of graph diffusion-based local clustering method on both the original and the weighted graphs. From a theoretical perspective, we consider recovering an unknown target cluster with a single seed node in a random graph with independent noisy node labels. We provide sufficient conditions on the label noise under which, with high probability, using diffusion in the weighted graph yields a more accurate recovery of the target cluster. This approach proves more effective than using the given labels alone or using diffusion in the label-free original graph. Empirically, we show that reliable node labels can be obtained with just a few samples from an attributed graph. Moreover, utilizing these labels via diffusion in the weighted graph leads to significantly better local clustering performance across several real-world datasets, improving F1 scores by up to 13%.

## 1 INTRODUCTION

Given a graph and a set of seed nodes from the graph, the task of local graph clustering aims to identify a small cluster of nodes that contains all or most of the seed nodes, without exploring the entire graph (Spielman & Teng, 2013; Orecchia & Zhu, 2014). Because of their ability to extract local structural properties within a graph and scalability to work with massive graphs, local graph clustering methods are frequently used in applications such as community detection, node ranking, and node embedding (Weng et al., 2010; Mahoney et al., 2012; Kloumann & Kleinberg, 2014; Perozzi et al., 2014; Gleich, 2015; Macgregor & Sun, 2021; Choromanski, 2023; Fountoulakis et al., 2023).

Traditionally, the problem of local graph clustering is studied under a simple, homogeneous context where the only available source of information is the connectivity of nodes, i.e. edges of the graph. There is often little hope to accurately identify a well-connected ground-truth target cluster which also has many external connections. Meanwhile, the emergence of heterogeneous data sources which consist of a graph and any additional node information like texts, images, or ground-truth labels offers new possibilities for improving existing clustering methods. This additional information can significantly benefit clustering, especially when the graph structure does not manifest a tightly-knit cluster of nodes. Yet, little effort has been made to formally investigate the benefits of combining multiple data sources for local graph clustering. Only recently, the work of Yang & Fountoulakis (2023) has studied the usage of node attributes under a strong homophily assumption. However, they require separable node attributes—often impractical for real-world data—and neglect other forms of node information that might be available. For example, in many cases, we also have access to the ground-truth labels of a small set of nodes which reveal their cluster affiliation, such as when it is known that certain nodes do not belong to the target cluster. While ground-truth label information has been extensively used in (semi-)supervised learning contexts and proved vital in numerous applications (Kipf & Welling, 2017; Hamilton et al., 2017), it is neither exploited by existing methods

for local graph clustering nor analyzed in proper theoretical settings with regard to how and why they can become helpful.

A challenge in designing local graph clustering algorithms that exploit additional sources of information and analyze their performance lies in the fact that additional data sources come in different forms. These can take the form of node or edge attributes, and even additional ground-truth labels for some nodes. Alternatively, one may just have access an oracle that outputs the likelihood of a node belonging to the target cluster, based on all available information. Due to the variability of attributed graph datasets in practice, a local graph clustering method and its analysis should ideally be agnostic to the specific source or form of additional information. To that end, in order to investigate the potential benefits that various additional sources of information can potentially bring to a local graph clustering task, we propose a study in the following setting.

***Local Graph Clustering with Noisy Node Labels:*** *Given a graph and a set of seed nodes, the goal is to recover an unknown target cluster around the seed nodes. Suppose that we additionally have access to noisy node labels (not to be confused with ground-truth labels), which are initially set to 1 if a node belongs to the target cluster and 0 otherwise, and then a fraction of them is flipped. How and when can these labels be used to improve clustering performance?*

In this context, node labels may be viewed as an abstract aggregation of all additional sources of information. The level of label noise, i.e. the fraction of flipped labels, controls the quality of the additional data sources we might have access to. From a practical point of view, noisy labels may be seen as the result of applying an imperfect classifier that predicts cluster affiliation of a node based on its attributes.[1] More generally, one may think of the noisy labels as the outputs of an encoder function that generates a binary label for a node based on all the additional sources of information we have for that node. The quality of both the encoder function and the data we have is thus represented by the label noise in the abstract setting that we consider.

Due to their wide range of and successful applications in practice (Mahoney et al., 2012; Kloumann & Kleinberg, 2014; Gleich, 2015; Eksombatchai et al., 2018; Fountoulakis et al., 2020), in this work, we focus on graph diffusion-based methods for local clustering. Our contributions are:

1. Given a graph $G$ and noisy node labels, we introduce a very simple yet surprisingly effective way to utilize the noisy labels for local graph clustering. We construct a weighted graph $G^w$ based on the labels and employ local graph diffusion in the weighted graph.
2. From a theoretical perspective, we analyze the performance of flow diffusion (Fountoulakis et al., 2020; Chen et al., 2022) over a random graph model, which is essentially a local (and more general) version of the stochastic block model. We focus on flow diffusion in our analysis due to its simplicity and good empirical performance (Fountoulakis et al., 2020; 2021). The diffusion dynamics of flow diffusion are similar to that of approximate personalized PageRank (Andersen et al., 2006) and truncated random walks (Spielman & Teng, 2013), and hence our results may be easily extended to other graph diffusions. We provide sufficient conditions on the label noise under which, with high probability, flow diffusion over the weighted graph $G^w$ leads to a more accurate recovery of the target cluster than flow diffusion over the original graph $G$.
3. We provide an extensive set of empirical experiments over 6 attributed real-world graphs, and we show that our method, which combines multiple sources of additional information, consistently leads to significantly better local clustering results than existing local methods. More specifically, we demonstrate that: (1) reasonably good node labels can be obtained as outputs of a classifier that takes as input the node attributes; (2) the classifier can be obtained with or without ground-truth label information, and it does not require access to the entire graph; (3) employing diffusion in the weighted graph $G^w$ outperforms both the classifier and diffusion in the original graph $G$.

## 1.1 RELATED WORK

The local graph clustering problem is first studied by Spielman & Teng (2013) using truncated random walks and by Andersen et al. (2006) using approximate personalized PageRank vectors. There is a long line of work on local graph clustering where the only available source of information is the graph

---

[1]The classifier may be obtained in a supervised manner in the presence of limited ground-truth label information, or in an unsupervised way without access to any ground-truth labels. If obtaining and applying such a classifier is part of a local clustering procedure, then neither its training nor inference should require full access to the graph. Our empirical results show that simple linear models can work surprisingly well in this context.

and a seed node (Spielman & Teng, 2013; Andersen et al., 2006; Chung, 2009; Reid & Yuval, 2009; Allen-Zhu et al., 2013; Andersen et al., 2016; Shi et al., 2017; Yin et al., 2017; Wang et al., 2017; Fountoulakis et al., 2020; Liu & Gleich, 2020). Most of existing local clustering methods are based on the idea of diffusing mass locally in the graph, including random walks (Spielman & Teng, 2013), heat diffusion (Chung, 2009), maximum flow (Wang et al., 2017) and network flow (Fountoulakis et al., 2020). We provide more details in Section 2 where we introduce the background in more depth.

Recently, Yang & Fountoulakis (2023) studies local graph clustering in the presence of node attributes. Under a strong assumption on the separability of node attributes, the authors provide upper bounds on the number of false positives when recovering a target cluster from a random graph model. The assumption requires that the Euclidean distance between every intra-cluster pair of node attributes has to be much smaller than the Euclidean distance between every inter-cluster pair of node attributes. Such an assumption may not hold in practice, for example, when the distributions of node attributes do not perfectly align with cluster affiliation, or when the node attributes are noisy. This restricts the practical effectiveness of their method. Our work takes a very different approach in that we do not restrict to a particular source of additional information or make any assumption on node attributes. Instead, we abstract all available sources of additional information as noisy node labels.

Leveraging multiple sources of information has been extensively explored in the context of graph clustering (Yang et al., 2013; Zhe et al., 2019; Sun et al., 2020), where one has access to both the graph and node attributes, and in the context of semi-supervised learning on graphs (Zhou et al., 2003; Kipf & Welling, 2017; Hamilton et al., 2017), where one additionally has access to some ground-truth class labels. All of these methods require processing the entire graph and all data points, and hence they are not suitable in the context of local graph clustering.

## 2 NOTATIONS AND BACKGROUND

We consider a connected, undirected, and unweighted graph $G = (V, E)$, where $V = \{1, 2, \ldots, n\}$ is a set of nodes and $E \subseteq V \times V$ is a set of edges. We focus on undirected and unweighted graphs for simplicity in our discussion, but the idea and results extend to weighted and strongly connected directed graphs. We write $i \sim j$ is $(i, j) \in E$ and denote $A \in \{0, 1\}^{n \times n}$ the adjacency matrix of $G$, i.e. $A_{ij} = 1$ is $i \sim j$ and $A_{ij} = 0$ otherwise. The degree of a node $i \in V$ is $\deg_G(i) := |\{j \in V : j \sim i\}|$, i.e. the number of nodes adjacent to it. The volume of a subset $U \subseteq V$ is $\mathrm{vol}_G(U) := \sum_{i \in U} \deg_G(i)$. We use subscripts to indicate the graph we are working with, and we omit them when the graph is clear from context. We write $E(U, W) := \{(i, j) \in E : i \in U, j \in W\}$ as the set of edges connecting two subsets $U, W \subseteq V$. The support of a vector $x \in \mathbb{R}^n$ is $\mathrm{supp}(x) := \{i : x_i \neq 0\}$. Throughout our discussion, we will denote $K$ as the target cluster we wish to recover, and write $K^c := V \backslash K$. Each node $i \in V$ is given a label $\tilde{y}_i \in \{0, 1\}$. For $c \in \{0, 1\}$ we write $\tilde{Y}_c := \{i \in V : \tilde{y}_i = c\}$. Throughout this work, labels that provide ground-truth information about cluster affiliation will be referred to as ground-truth labels. When the word *labels* is mentioned without the modifier *ground-truth*, one should interpret those as the noisy labels $\tilde{y}_i$.

In the traditional setting for local graph clustering, we are given a seed node $s$ which belongs to an unknown target cluster $K$, or sometimes more generally, we are given a set of seed nodes $S \subset V$ such that $S \cap K \neq \emptyset$. It is often assumed that the size of the target cluster $K$ is much smaller than the size of the graph, and hence a good local clustering method should ideally be able to recover $K$ without having to explore the entire graph. In fact, the running times of nearly all existing local clustering algorithms scale only with the size of $K$ instead of the size of the graph (Spielman & Teng, 2013; Andersen et al., 2006; 2016; Wang et al., 2017; Fountoulakis et al., 2020; Martínez-Rubio et al., 2023). This distinguishes the problem of local graph clustering from other problems which require processing the entire graph. In order to obtain a good cluster around the seed nodes, various computational routines have been tested to locally explore the graph structure. One of the most widely used ideas is local graph diffusion. Broadly speaking, local graph diffusion is a process of spreading certain mass from the seed nodes to nearby nodes along the edges of the graph. For example, approximate personalized PageRank iteratively spreads probability mass from a node to its neighbors (Andersen et al., 2006), heat kernel PageRank diffuses heat from the seed nodes to the rest of the graph (Chung, 2009), capacity releasing diffusion spreads source mass by following a combinatorial push-relabel procedure (Wang et al., 2017), and flow diffusion routes excess mass out of the seed nodes while minimizing a network flow cost (Fountoulakis et al., 2020). In a local graph diffusion process, mass tends to spread within well-connected clusters, and hence a careful look at where mass spreads to in the graph often yields a good local clustering result.

Our primary focus is the flow diffusion (Fountoulakis et al., 2020) and in particular its $\ell_2$-norm version (Chen et al., 2022). We choose flow diffusion due to its good empirical performance and its flexibility in initializing a diffusion process. The flexibility allows us to derive results that are directly comparable with the accuracy of given noisy labels. In what follows, we provide a brief overview of the $\ell_2$-norm flow diffusion. For a more in-depth introduction, we refer the readers to Fountoulakis et al. (2020) and Chen et al. (2022). In a flow diffusion, we are given $\Delta_s > 0$ units of source mass for each seed node $s$. Every node $i \in V$ has a capacity, $T_i \geq 0$ which is the maximum amount of mass it can hold. If $\Delta_i > T_i$ at some node $i$, then we need to spread mass from $i$ to its neighbors in order to satisfy the capacity constraint. Flow diffusion spreads mass along the edges in a way such that the $\ell_2$-norm of mass that is passed over the edges is minimized. In particular, given a weight vector $w \in \mathbb{R}^{|E|}$ (i.e., edge $e$ has resistance $1/w_e$), the $\ell_2$-norm flow diffusion and its dual can be formulated as follows (Chen et al., 2022):

$$\min \ \frac{1}{2} \sum_{e \in E} f_e^2 / w_e \quad \text{s.t. } B^T f \leq T - \Delta, \tag{1}$$

$$\min \ x^T L x + x^T (T - \Delta) \quad \text{s.t. } x \geq 0, \tag{2}$$

where $B \in \mathbb{R}^{|E| \times |V|}$ is the signed edge incidence matrix under an arbitrary orientation of the graph, and $L = B^T W B$ is the (weighted) graph Laplacian matrix where $W$ is the diagonal matrix of $w$. If the graph is unweighted, then we treat $w$ as the all-ones vector. In the special case where we set $\Delta_s = 1$ for some node $s$ and 0 otherwise, $T_t = 1$ for some node $t$ and 0 otherwise, the flow diffusion problem (1) reduces to an instance of electrical flows (Christiano et al., 2011). Electrical flows are closely related to random walks and effective resistances, which are useful for finding good global cuts that partition the graph. In a flow diffusion process, one has the flexibility to choose source mass $\Delta$ and sink capacity $T$ so that the entire process only touches a small subset of the graph. For example, if $T_i = 1$ for all $i$, then one can show that the optimal solution $x^*$ for the dual problem (2) satisfies $|\text{supp}(x^*)| \leq \sum_{i \in V} \Delta_i$, and moreover, the solution $x^*$ can be obtained in time $O(|\text{supp}(x^*)|)$ which is independent of either $|V|$ or $|E|$ (Fountoulakis et al., 2020). This makes flow diffusion useful in the local clustering context. The solution $x^*$ provides a scalar embedding for each node in the graph. With properly chosen $\Delta$ and $T$, Fountoulakis et al. (2020) showed that applying a sweep procedure on the entries of $x^*$ returns a low conductance cluster, and Yang & Fountoulakis (2023) showed that $\text{supp}(x^*)$ overlaps well with an unknown target cluster in a contextual random graph model with very informative node attributes.

## 3 Label-based edge weights improve clustering accuracy

In this section, we discuss the problem of local graph clustering with noisy node labels. Given a graph $G = (V, E)$ and noisy node labels $\tilde{y}_i \in \{0, 1\}$ for $i \in V$, the goal is to identify an unknown target cluster $K$ around a set of seed nodes. We primarily focus on using the $\ell_2$-norm flow diffusion (2) for local clustering, but our method and results should easily extend to other diffusion methods such as approximate personalized PageRank.[2] Let $x^*$ denote the optimal solution of (2), we adopt the same rounding strategy as Ha et al. (2021) and Yang & Fountoulakis (2023), that is, we consider $\text{supp}(x^*)$ as the output cluster and compare it against the target $K$. Specific diffusion setup with regard to the source mass $\Delta$ and sink capacity $T$ is discussed in Section 3.1. Occasionally we write $x^*(T, \Delta)$ or $x^*(\Delta)$ to emphasize its dependence on the source mass and sink capacity, and we omit them when they are clear from the context. Whenever deemed necessary for the sake of clarity, we use different superscripts $x^*$ and $x^\dagger$ to distinguish solutions of (2) obtained under different edge weights.

We will denote

$$a_1 := |K \cap \tilde{Y}_1| / |K|, \quad a_0 := |K^{\mathsf{c}} \cap \tilde{Y}_0| / |K^{\mathsf{c}}|, \tag{3}$$

which quantify the accuracy of labels within $K$ and outside $K$, respectively. If $a_0 = a_1 = 1$, then the labels perfectly align with cluster affiliation. We say that the labels are noisy if at least one of $a_0$ and $a_1$ is strictly less than 1. In this case, we are interested in how and when the labels can help local graph diffusion obtain a more accurate recovery of the target cluster. In order for the labels to provide any useful information at all, we will assume that the accuracy of these labels is at least 1/2.

**Assumption 3.1.** The label accuracy satisfies $a_0 \geq 1/2$ and $a_1 \geq 1/2$.

---

[2] We demonstrate this through a comprehensive empirical study in Appendix D.2.

To exploit the information provided by noisy node labels, we consider a straightforward way to weight the edges of the graph $G = (V, E)$ based on the given labels. Denote $G^w = (V, E, w)$ the weighted graph obtained by assigning edge weights according to $w : E \to \mathbb{R}$. We set

$$w(i, j) = 1 \text{ if } \tilde{y}_i = \tilde{y}_j, \quad \text{and} \quad w(i, j) = \epsilon \text{ if } \tilde{y}_i \neq \tilde{y}_j, \tag{4}$$

for some small $\epsilon \in [0, 1)$. This assigns a small weight over cross-label edges and maintains unit weight over same-label edges. The value of $\epsilon$ interpolates between two special scenarios. If $\epsilon = 1$ then $G^w$ reduces to $G$, and if $\epsilon = 0$ then all edges between $\tilde{Y}_1$ and $\tilde{Y}_0$ are removed in $G^w$. In principle, the latter case can be very helpful if the labels are reasonably accurate or the target cluster is well-connected. For example, in Section 3.1 we will show that solving the $\ell_2$-norm flow diffusion problem (2) over the weighted graph $G^w$ by setting $\epsilon = 0$ can lead to a more accurate recovery of the target cluster than solving it over the original graph $G$. In practice, one may also choose a small nonzero $\epsilon$ to improve robustness against very noisy labels. In our experiments, we find that the results are not sensitive to the choice of $\epsilon$, as we obtain similar clustering accuracy for $\epsilon \in [10^{-2}, 10^{-1}]$ over different datasets with varying cluster sizes and label accuracy. We use the F1 score to measure the accuracy of cluster recovery. Suppose that a local clustering algorithm returns a cluster $C \subset V$, then

$$\text{F1}(C) = \frac{|K|}{|K| + |C \backslash K|/2 + |K \backslash C|/2}.$$

Even though the edge weights (4) are fairly simple and straightforward, they lead to surprisingly good local clustering results over real-world data, as we will show in Section 4. Before we formally analyze diffusion over $G^w$, let us start with an informal and intuitive discussion on how such edge weights can be beneficial. Consider a step during a generic local diffusion process where mass is spread from a node within the target cluster to its neighbors. Suppose this node has a similar number of neighbors within and outside the target cluster. An illustrative example is shown in Figure 1a. In this case, since all edges are treated equally, diffusion will spread a lot of mass to the outside. This makes it very difficult to accurately identify the target cluster without suffering from excessive false positives and false negatives. On the other hand, if the labels have good initial accuracy, for example, if $a_1 > a_0$, then weighting the edges according to (4) will generally make more boundary edges smaller while not affecting as many internal edges. This is illustrated in Figure 1b. Since a diffusion step spreads mass proportionally to the edge weights (Xing & Ghorbani, 2004; Xie et al., 2015; Yang & Fountoulakis, 2023), a neighbor that is connected via a lower edge weight will receive less mass than a neighbor that is connected via a higher edge weight. Consequently, diffusion in such a weighted setting forces more mass to be spread within the target cluster, and hence less mass will leak out. This generally leads to a more accurate recovery of the target cluster.

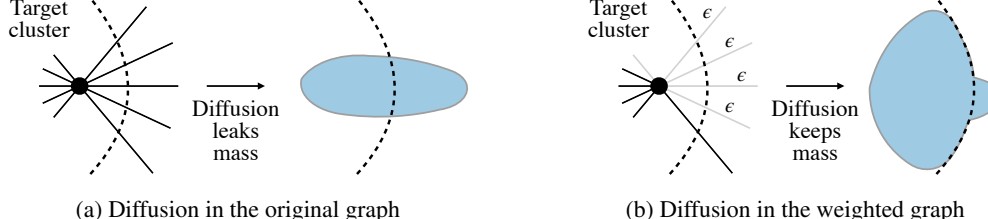

(a) Diffusion in the original graph       (b) Diffusion in the weighted graph

Figure 1: Label-based edge weights avoid mass leakage by attenuating more boundary edges than internal edges. This helps local diffusion more accurately recover the target cluster.

## 3.1 Guaranteed improvement under a random graph model

Following prior work on statistical analysis of local graph clustering algorithms (Ha et al., 2021; Yang & Fountoulakis, 2023), we assume that the graph and the target cluster are generated from the following random model, which can be seen as a localized stochastic block model.

**Definition 3.2.** [Local random model (Ha et al., 2021; Yang & Fountoulakis, 2023)] Given a set of nodes $V$ and a target cluster $K \subset V$. For every pair of nodes $i$ and $j$, if $i, j \in K$ then we draw an edge $(i, j)$ with probability $p$; if $i \in K$ and $j \in K^{\mathsf{c}}$ then we draw an edge $(i, j)$ with probability $q$; otherwise, if both $i, j \in K^{\mathsf{c}}$ then we allow any (deterministic or random) model to draw an edge.

Let $k = |K|$ and $n = |V|$. For a node $i \in K$, the expected number of internal connections is $p(k-1)$ and the expected number of external connections is $q(n-k)$. We will denote their ratio by

$$\gamma := \frac{p(k-1)}{q(n-k)}.$$

The value of $\gamma$ can be seen as a measure of the structural signal of the target cluster $K$. When $\gamma$ is small, a node within $K$ is likely to have more external connections than internal connections; when $\gamma$ is large, a node within $K$ is likely to have more internal connections than external connections.

Recall our definition of the weighted graph $G^w$ whose edge weights are given in (4) based on node labels. We will study the effect of incorporating noisy labels by comparing the clustering accuracy obtained by solving the flow diffusion problem (2) over the original graph $G$ and the weighted graph $G^w$, respectively. For the purpose of the analysis, we simply set $\epsilon = 0$ as this is enough to demonstrate the advantage of diffusion over $G^w$. Determining an optimal and potentially nonzero $\epsilon \in [0, 1]$ that maximizes accuracy is an interesting question and we provide additional discussion in Appendix B. For readers familiar with the literature on local clustering by minimizing conductance, if $p$ and $q$ are reasonably large, e.g., $p \geq 4 \log k / k$ and $q \geq 4 \log k / (n-k)$, then a simple computation invoking the Chernoff bound yields that

$$\text{cut}_{G^w}(K) \asymp (a_1(1-a_0) + a_0(1-a_1)) \cdot \text{cut}_G(K), \quad \text{with high probability,}$$

$$\text{vol}_{G^w[K]}(K) \asymp (a_1^2 + (1-a_1)^2) \cdot \text{vol}_{G[K]}(K), \quad \text{with high probability,}$$

where $G[C]$ denotes the subgraph induced on $C \subseteq V$, so $\text{vol}_{G[K]}(K)$ measures the overall internal connectivity of $K$. Since $a_1(1-a_0) + a_0(1-a_1) < a_1^2 + (1-a_1)^2$ as long as $a_0, a_1 > 1/2$, the target cluster $K$ will have a smaller conductance in $G^w$ as long as the label accuracy is larger than $1/2$. As a result, this potentially improves the detectability of $K$ in $G^w$. Of course, a formal argument requires careful treatments of diffusion dynamics in $G$ and $G^w$, respectively.

We consider local clustering with a single seed node using flow diffusion processes where the sink capacity is set to $T_i = 1$ for all $i \in V$. Although discussed earlier, we summarize below the key steps of local clustering with noisy labels using flow diffusion:

**Input:** Graph $G = (V, E)$, seed node $s \in V$, noisy labels $\tilde{y}_i$ for $i \in V$, source mass parameter $\theta$.
1. Create weighted graph $G^w$ based on (4). Set source mass $\Delta_i = \theta$ if $i = s$ and 0 otherwise.
2. Solve the $\ell_2$-norm flow diffusion problem (2) over $G^w$. Obtain solution $x^\dagger(\theta)$.
3. Return a cluster $C = \text{supp}(x^\dagger(\theta))$.

*Remark* 3.3 (Locality). In a practical implementation of the method, Step 1 and Step 2 can be carried out without accessing the full graph. This is because computing the solution $x^\dagger$ only requires access to nodes (and their labels) that either belong to $\text{supp}(x^\dagger)$ or are neighbors of a node in $\text{supp}(x^\dagger)$. See, for example, Algorithm 1 from Yang & Fountoulakis (2023) which provides a local algorithm for solving the $\ell_2$-norm flow diffusion problem (2). Recall that the amount of source mass controls the size of $\text{supp}(x^\dagger)$. In the above setup, $\theta$ controls the amount of source mass, and since $T_i = 1$ for all $i$, we have $|\text{supp}(x^\dagger)| \leq \theta$. Applying the complexity results in Fountoulakis et al. (2020); Yang & Fountoulakis (2023), we immediately get that the total running time of the above steps are $O(\bar{d}\theta)$ where $\bar{d}$ is the maximum node degree in $\text{supp}(x^*)$. This makes the running time independent of $|V|$.

Given source mass $\theta \geq 0$ at the seed node, let $x^*(\theta)$ and $x^\dagger(\theta)$ denote the solutions obtained from solving (2) over $G$ and $G^w$, respectively. Theorem 3.4 provides a lower bound on the F1 score obtained by $\text{supp}(x^\dagger(\theta^\dagger))$ with appropriately chosen source mass $\theta^\dagger$. In addition, it gives a sufficient condition on the label accuracy $a_0$ and $a_1$ such that flow diffusion over $G^w$ with source mass $\theta^\dagger$ at the seed node results in a more accurate recovery of $K$ than flow diffusion over $G$ with any possible choice of source mass. For the sake of simplicity in presentation, Theorem 3.4 has been simplified from the long and more formal version provided in Appendix A (see Theorem A.1). The long version requires weaker assumptions on $p, q$ and provides exact terms without involving asymptotics.

**Theorem 3.4** (Simplified version). *Suppose that $p = \omega(\frac{\sqrt{\log k}}{\sqrt{k}})$ and $q = \omega(\frac{\log k}{n-k})$. With probability at least $1 - 1/k$, there is a set $K' \subseteq K$ with cardinality at least $|K|/2$ and a choice of source mass $\theta^\dagger$, such that for every seed node $s \in K'$ we have*

$$\text{F1}(\text{supp}(x^\dagger(\theta^\dagger))) \geq \left[ 1 + \frac{(1-a_1)}{2} + \frac{(1-a_0)}{2\gamma} + \frac{(1-a_0)^2}{2a_1\gamma^2} \right]^{-1} - o_k(1). \tag{5}$$

*Furthermore, if the accuracy of noisy labels satisfies*

$$a_0 \geq 1 - \left( \sqrt{(p/\gamma + 2a_1 - 1)\, a_1} - a_1 \right) \gamma + o_k(1), \tag{6}$$

*then we have* $\mathrm{F1}(\mathrm{supp}(x^\dagger(\theta^\dagger))) > \max_{\theta \geq 0} \mathrm{F1}(\mathrm{supp}(x^*(\theta)))$.

Theorem 3.4 requires additional discussion. First, the lower bound in (5) increases with $a_0$, $a_1$ and $\gamma$. This naturally corresponds to the intuition that, as the label accuracy $a_0$ and $a_1$ become larger, we can expect a more accurate recovery of $K$; while on the other hand, as the local graph structure becomes noisy, i.e. as $\gamma$ becomes smaller, it generally becomes more difficult to accurately recover $K$. Note that when $a_0 = a_1 = 1$, the labels perfectly align with cluster affiliation, and in this special case the lower bound on the F1 score naturally becomes $1 - o_k(1)$. This means that the solution $x^\dagger$ over the weighted graph $G^w$ fully leverages the perfect label information. Finally, notice from (5) that the F1 is lower bounded by a constant as long as $\gamma = \Omega_n(1)$, even if $a_0$ is as low as $1/2$. In comparison, under a typical local clustering context where $k \ll n$, the F1 score obtained from directly using the noisy labels can be arbitrarily close to 0, i.e. we have $\mathrm{F1}(\tilde{Y}_1) \leq o_n(1)$, as long as $a_0$ is bounded away from 1. This demonstrates the importance of employing local diffusion. Even when the initial labels are deemed fairly accurate based on $a_0$ and $a_1$, e.g. $a_1 = 1$, $a_0 = 0.99$, the F1 score of the labels can still be very low. In the next section, over both synthetic and real-world data, we show empirically that flow diffusion over the weighted graph $G^w$ can result in surprisingly better F1 even when the F1 of labels is very poor.

Second, if $a_1 = 1$ then (6) becomes

$$a_0 \geq 1 - \left( \sqrt{1 + p/\gamma} - 1 \right) \gamma + o_k(1). \tag{7}$$

Observe that: (i) The function $\sqrt{\gamma^2 + p\gamma} - \gamma$ is increasing with $\gamma$, therefore the left-hand side of (7) increases as $\gamma$ decreases. This corresponds to the intuition that, as the external connectivity of the target cluster becomes larger (i.e. as $\gamma$ decreases), we need more accurate labels to prevent a lot of mass from leaking out. (ii) When $q \geq \Omega(\frac{k}{n-k})$, we have $p/\gamma \geq \Omega(1)$, and (7) further simplifies to $a_0 \geq 1 - \Omega(\gamma)$. In this case, if $\gamma$ is also constant, we can expect that flow diffusion over $G^w$ to give a better result even if a constant fraction of labels is incorrect. Here, the required conditions on $q$ and $\gamma$ may look a bit strong because we did not assume anything about the graph structure outside $K$. One may obtain much weaker conditions than (6) or (7) under additional assumptions on $K^c$.

## 4 EXPERIMENTS

In this section, we evaluate the effectiveness of employing flow diffusion over the label-weighted graph $G^w$ whose edge weights are given in (4) for local clustering. We will refer to it as Label-based Flow Diffusion (**LFD**). We compare the results with the standard $\ell_2$-norm flow diffusion (**FD**) (Fountoulakis et al., 2020; Chen et al., 2022). Whenever a dataset includes node attributes, we also compare with the weighted flow diffusion (**WFD**) from Yang & Fountoulakis (2023). Due to space constraints, we only report experiments involving flow diffusion in the main paper. We carried out extensive experiments comparing Label-based PageRank (LPR) on the weighted graph $G^w$ with PageRank (PR) on the original graph $G$. The results are similar: LPR consistently outperforms PR. Experiments and results that involve PageRank can be found in Appendix D. In addition, we show that our method is not very sensitive to hyperparameter choice (see Appendix C.1) and that our method maintains the fast running time of traditional local diffusion algorithms (see Appendix C.2).

We use both synthetic and real-world data to evaluate the methods. The synthetic data is used to demonstrate our theory and show how local clustering performance improves as label accuracy increases in a controlled environment. For the real-world data, we consider both supervised (i.e. we have access to both node attributes and some ground-truth labels) and unsupervised (i.e. we have access to only node attributes) settings. We show that in both settings, one may easily obtain reasonably good node labels such that, leveraging these labels via diffusion over $G^w$ leads to consistently better results across all 6 datasets, improving the F1 score by up to 13%.

### 4.1 EXPERIMENTS ON SYNTHETIC DATA

We generate a synthetic graph using the stochastic block model with cluster size $k = 500$ and number of clusters $c = 20$. The number of nodes in the graph equals $n = kc = 10,000$. Two nodes within

the same cluster are connected with probability $p = 0.05$, and two nodes from different clusters are connected with probability $q = 0.025$. For edge weights (4) in $G^w$ we set $\epsilon = 0.05$. We vary the label accuracy $a_0$ and $a_1$ as defined in (3) to demonstrate the effects of varying label noise. We consider 3 settings: (i) fix $a_0 = 0.7$ and vary $a_1 \in [1/2, 1)$; (ii) fix $a_1 = 0.7$ and vary $a_0 \in [1/2, 1)$; (iii) vary both $a_0, a_1 \in [1/2, 1)$ at the same time. For each pair of $(a_0, a_1)$, we run 100 trials. For each trial, we randomly select one of the 20 clusters as the target cluster. Then we generate noisy labels according to $a_0$ and $a_1$. For each trial, we randomly select a node from the target cluster as the seed node. We set the sink capacity $T_i = 1$ for all nodes. For the source mass at the seed node, we set it to $\alpha k$ for $\alpha = 2, 2.25, \ldots, 4$, out of which we select the one that results in the highest F1 score based on $\mathrm{supp}(x^*)$, where $x^*$ is the optimal solution of the flow diffusion problem (2).[3]

We compare the F1 scores achieved by FD and LFD over varying levels of label accuracy. Recall that FD does not use and hence is not affected by the labels at all, whereas LFD uses label-based edge weights from (4). The results of over 100 trials are shown in Figure 2. In addition to the results obtained from flow diffusion, we also include the F1 scores obtained from the labels alone (**Labels**), i.e. we compare $\tilde{Y}_1 = \{i \in V : \tilde{y}_i = 1\}$ against $K$. Not surprisingly, as predicted by (5), the F1 of LFD increases as at least one of $a_0, a_1$ increases. Moreover, LFD already outperforms FD at reasonably low label accuracy, e.g. when $a_0, a_1 = 0.7$ and the F1 of the labels alone is as low as 0.2. This shows the effectiveness of incorporating noisy labels and employing diffusion over the label-weighted graph $G^w$. Even fairly noisy node labels can boost local clustering performance.

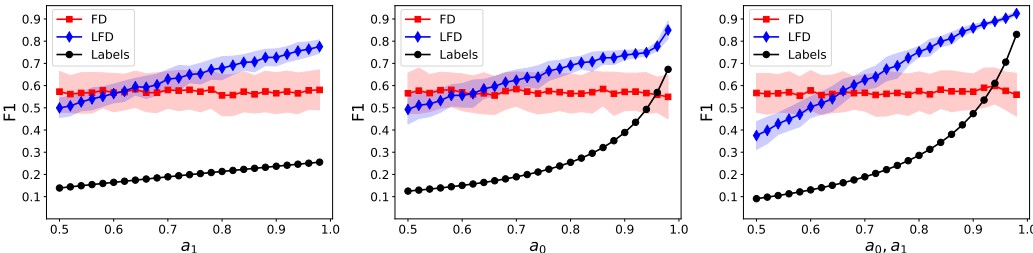

Figure 2: F1 scores obtained by employing flow diffusion over the original graph (FD) and the label-weighted graph (LFD). For comparison, we also plot the F1 obtained by the noisy labels (Labels). The solid line and error bar show mean and standard deviation over 100 trials, respectively. As discussed in Section 3.1, even fairly noisy labels can already help boost local clustering performance.

## 4.2 EXPERIMENTS ON REAL-WORLD DATA

We carry out experiments over the following 6 real-world attributed graph datasets. We include all 3 datasets used in Yang & Fountoulakis (2023)—namely, Amazon Photo (McAuley et al., 2015), Coauthor CS, and Coauthor Physics (Shchur et al., 2018)—to ensure compatibility of results. Additionally, we use 3 well-established graph machine learning benchmarks: Amazon Computers (Shchur et al., 2018), Cora (McCallum et al., 2000) and Pubmed (Sen et al., 2008). We provide the most informative results in this section. Detailed empirical setup and additional results are found in Appendix D.

We divide the experiments into two settings. In the first, we assume access to a selected number of ground-truth labels, evenly sampled from both the target and non-target classes. These nodes are utilized to train a classifier (without graph information). The predictions of the classifier are then used as noisy labels to construct the weighted graph $G^w$ as defined in (4), and we set $\epsilon = 0.05$ as in the synthetic experiments. We use all the positive nodes, i.e. nodes that belong to the target cluster based on the given ground-truth labels, as seed nodes during the diffusion process. For each cluster in each dataset, we compare LFD against FD and WFD over 100 trials. For each trial, a classifier is trained using randomly sampled positive and negative nodes which we treat as ground-truth information. Figure 3 shows the average F1 obtained by each method versus the number of samples used for training the classifier. As illustrated in Figure 3, using the outputs from a weak classifier (e.g. with an F1 score as low as 40%) as noisy labels already enhances the diffusion process, obtaining an improvement as high as 13% over other methods (see Coauthor Physics with 25 positive and negative

---

[3]We do this for our synthetic experiments to illustrate how label accuracy affects local clustering performance. In practice, without the ground-truth information, one may fix a reasonable $\alpha$, e.g. $\alpha = 2$, and then apply a sweep-cut procedure on $x^*$. We adopt the latter approach for experiments on real-world data.

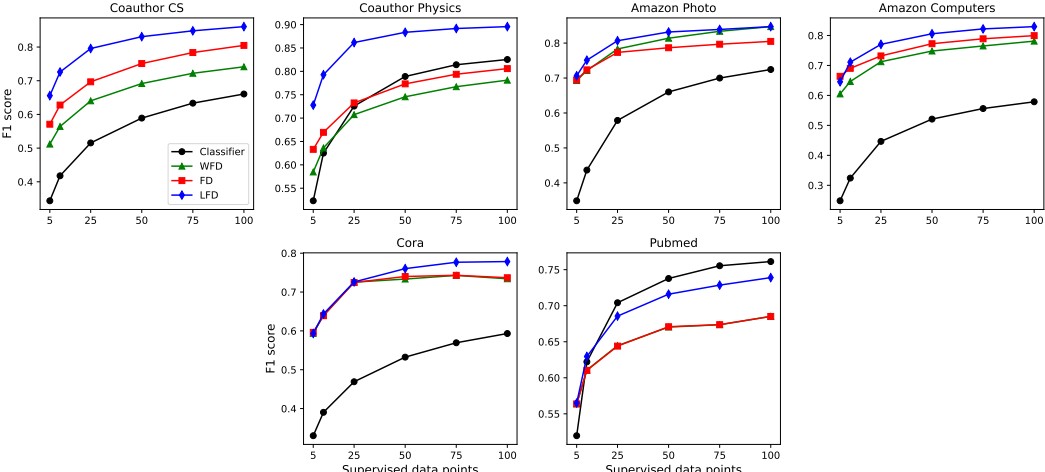

Figure 3: F1 scores for local clustering using Flow Diffusion (FD), Weighted Flow Diffusion (WFD), Label-based Flow Diffusion (LFD), and Logistic Regression (Classifier) with an increasing number of positive and negative ground-truth samples.

nodes). The increasing availability of ground-truth positive nodes typically benefits all diffusion processes. However, as seen in Cora, additional seed nodes can also increase the risk of mass leakage outside the target class and hence result in lower accuracy. In such cases, the learned classifier mitigates this problem by reducing inter-edge weights.

In the second set of experiments, we consider the setting where we are only given a single seed with no access to ground-truth labels or a pre-trained classifier. To demonstrate the effectiveness of our method, a heuristic approach is adopted. First, we solve the flow diffusion problem (2) over the original graph $G$ and get a solution $x^*$. Then, we select 100 nodes with the highest and lowest values in $x^*$, which are designated as positive and negative nodes, respectively. We use these nodes to train a binary classifier. As demonstrated in prior work (Fountoulakis et al., 2020; Yang & Fountoulakis, 2023), nodes with the highest values in $x^*$ typically belong to the target cluster, whereas nodes with the lowest values in $x^*$ — typically zero — are outside of the target cluster. We use the outputs of the classifier as noisy node labels to construct the weighted graph $G^w$. We test this approach against the standard and weighted flow diffusion, both in the single and multi-seed settings. In the multi-seed setting, the 100 (pseudo-)positive nodes are used as seed nodes. Additionally, for each dataset, we compare LFD with the best-performing baseline and report the improvement in Table 1. The results demonstrate a consistent improvement of LFD over other methods across all datasets.

Table 1: Comparison of F1 scores across datasets for Flow Diffusion (FD), Weighted Flow Diffusion (WFD), and Label-based Flow Diffusion (LFD) in the absence of ground-truth information

| Dataset | FD (single-seed) | WFD (single-seed) | FD (multi-seed) | WFD (multi-seed) | LFD | Improv. ($\pm$) | Improv. (%) |
|---|---|---|---|---|---|---|---|
| Coauthor CS | 43.8 | 39.9 | 50.5 | 47.1 | **63.1** | +12.6 | +24.9 |
| Coauthor Physics | 62.8 | 57.0 | 55.5 | 51.1 | **72.9** | +10.1 | +16.1 |
| Amazon Photo | 54.5 | 57.4 | 62.1 | 62.6 | **66.8** | +4.2 | +6.7 |
| Amazon Computers | 56.2 | 53.3 | 58.2 | 54.6 | **60.4** | +2.2 | +3.8 |
| Cora | 33.3 | 33.7 | 55.4 | 55.4 | **56.5** | +1.1 | +1.9 |
| Pubmed | 53.0 | 53.2 | 53.9 | 53.9 | **55.3** | +1.4 | +2.7 |
| AVERAGE | 50.6 | 49.1 | 55.9 | 54.1 | 62.5 | +5.3 | +9.3 |

## 5 CONCLUSION

We introduce the problem of local graph clustering with access to noisy node labels. This new problem setting serves as a proxy for working with real-world graph data with additional node information. Moreover, such setting allows for developing local methods that are agnostic to the actual sources and formats of additional information which can vary from case to case. We propose a simple label-based edge weight scheme to utilize the noisy labels, and we show that performing local clustering over the weighted graph is effective both in theory and in practice.

ACKNOWLEDGMENTS

K. Fountoulakis would like to acknowledge the support of the Natural Sciences and Engineering Research Council of Canada (NSERC). Cette recherche a été financée par le Conseil de recherches en sciences naturelles et en génie du Canada (CRSNG), [RGPIN-2019-04067, DGECR-2019-00147].

S. Yang would like to acknowledge the support of the Natural Sciences and Engineering Research Council of Canada (NSERC).

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

## A  FORMAL STATEMENT OF THEOREM 3.4 AND PROOFS

For convenience let us remind the reader the notations that we use. We use $K \subset V$ to denote the target cluster and we write $K^{\mathsf{c}} := V \setminus K$. Each node $i \in V$ comes with a noisy label $\tilde{y}_i \in \{0, 1\}$. For $c \in \{0, 1\}$ we write $\tilde{Y}_c := \{i \in V : \tilde{y}_i = c\}$. The label accuracy are characterized by $a_0 = |K^{\mathsf{c}} \cap \tilde{Y}_0|/|K^{\mathsf{c}}|$ and $a_1 = |K \cap \tilde{Y}_1|/|K|$. Given a graph $G = (V, E)$ and a target cluster $K$ generated by the random model in Definition 3.2, let $n := |V|$, $k := |K|$, and $\gamma := \frac{p(k-1)}{q(n-k)}$. Given edge weight $w : E \to \mathbb{R}_+$ or equivalently a vector $w \in \mathbb{R}_+^{|E|}$, let $G^w$ denote the weighted graph obtained by assigning edge weights to $G$ according to $w$. In our analysis we consider edge weights given by (4), that is $w(i, j) = 1$ if $\tilde{y}_i = \tilde{y}_j$, and $w(i, j) = \epsilon$ if $\tilde{y}_i \neq \tilde{y}_j$, and we set $\epsilon = 0$.

Recall that the $\ell_2$-norm flow diffusion problem (2) is set up as follows. The sink capacity is set to $T_i = 1$ for all $i \in V$. We set $T_i = 1$ instead of $T_i = \deg_G(i)$ as used in Fountoulakis et al. (2020) because it allows us to derive bounds on the F1 score in a more direct way. In practice, both can be good choices. For a given seed node $s \in K$, we set source mass $\Delta_s = \theta$ at node $s$ for some $\theta > 0$, and we set $\Delta_i = 0$ for all other nodes $i \neq s$. Given source mass $\theta$ at the seed node, let $x^*(\theta)$ and $x^\dagger(\theta)$ denote the solutions of the $\ell_2$-norm flow diffusion problem (2) over $G$ and $G^w$, respectively. We write $x^*(\theta)$ and $x^\dagger(\theta)$ to emphasize their dependence on $\theta$. When the choice of $\theta$ is clear from the context, we simply write $x^*$ and $x^\dagger$.

We state the formal version of Theorem 3.4 below in Theorem A.1. First, let us define two numeric quantities. Given $0 < \delta_1, \delta_2, \delta_3 \leq 1$, let

$$r := \frac{(1 + \delta_1)(1 + \delta_1 + \frac{2}{p(k-1)})}{(1 - \delta_1)(1 - \delta_2))}, \quad \text{and} \quad r' := r/(1 - \delta_3).$$

**Theorem A.1** (Formal version of Theorem 3.4). *Suppose that* $p \geq \max\left(\frac{(6+\epsilon_1)}{\delta_1^2} \frac{\log k}{k-2}, \frac{(\sqrt{8}+\epsilon_2)}{\delta_2\sqrt{1-\delta_1}} \frac{\sqrt{\log k}}{\sqrt{k-2}}\right)$ *and* $q \geq \frac{(3+\epsilon_3)}{\delta_3^2} \frac{\log k}{n-k}$ *for some* $\epsilon_1, \epsilon_2, \epsilon_3 > 0$ *and* $0 < \delta_1, \delta_2, \delta_3 \leq 1$. *Then with probability at least* $1 - 3k^{-\epsilon_1/6} - k^{-\epsilon_2} - k^{-\epsilon_3/3}$, *there is a set* $K' \subseteq K$ *with cardinality at least* $|K|/2$ *and a choice of source mass* $\theta^\dagger$, *such that for every seed node* $s \in K'$ *with source mass* $\theta^\dagger$ *at the seed node, we get*

$$\mathrm{F1}(\mathrm{supp}(x^\dagger(\theta^\dagger))) \geq \left[1 + \frac{a_1}{2}\left(\left(\frac{a_1\gamma\frac{(k-2)}{(k-1)} + (1-a_0)}{a_1\gamma\frac{(k-2)}{(k-1)}}\right)^2 r - 1\right) + \frac{1-a_1}{2}\right]^{-1}. \quad (8)$$

*In this case, if the accuracy of noisy labels satisfies,*

$$a_0 \geq 1 - \frac{(k-2)}{(k-1)}\left(\sqrt{\left(\frac{p/\gamma}{r'} + \frac{2a_1 - 1}{r}\right)a_1} - a_1\right)\gamma, \quad (9)$$

*then we have*

$$\mathrm{F1}(\mathrm{supp}(x^\dagger(\theta^\dagger))) > \max_{\theta \geq 0} \mathrm{F1}(\mathrm{supp}(x^*(\theta))).$$

**Outline:** The proof is based on (1) lower bounding the number of false positives incurred by $\mathrm{supp}(x^*)$ (Proposition A.2), (2) upper bounding the number of false positives incurred by $\mathrm{supp}(x^\dagger)$ (Proposition A.3), and (3) combine both lower and upper bounds.

We will use some concentration results concerning the connectivity of the random graphs $G$ and $G^w$. These results are mostly derived from straightforward applications of the Chernoff bound. For completeness we state these results and provide their proofs at the end of this section.

Let $\hat{x}$ denote a generic optimal solution of the flow diffusion problem (2), which is obtained over either $G$ or $G^w$. We will heavily use the following two important properties of $\hat{x}$ (along with its physical interpretation). We refer the reader to Fountoulakis et al. (2020) for details.

1. The solution $\hat{x} \in \mathbb{R}^n$ defines a flow diffusion over the underlying graph such that, for all $i, j \in V$, the amount of mass that node $i$ sends to node $j$ along eege $(i, j)$ is given by $w(i, j) \cdot (\hat{x}_i - \hat{x}_j)$.

2. For all $i \in V$, $\hat{x}_i > 0$ only if the total amount of mass that node $i$ has equals $T_i$, i.e., $\Delta_i + \sum_{j \sim i} w(i, j) \cdot (\hat{x}_j - \hat{x}_i) = T_i$.

3. For all $i \in V$, $\hat{x}_i > 0$ if and only if the total amount of mass that node $i$ receives exceeds $T_i$, i.e.,
$\Delta_i + \sum_{j \sim i, \hat{x}_j > \hat{x}_i} w(i,j) \cdot (\hat{x}_j - \hat{x}_i) > T_i$.

Using these properties we may easily obtain a lower bound on the number of false positives incurred by $\mathrm{supp}(x^*)$. Recall that $x^*$ denotes the optimal solution of (2) over $G$, with sink capacity $T_i = 1$ for all $i$. We state the result in Proposition A.2.

**Proposition A.2.** *If $q \geq \frac{(3+\epsilon)}{\delta^2} \frac{\log k}{n-k}$ for some $\epsilon > 0$ and $0 < \delta \leq 1$, then with probability at least $1 - k^{-\epsilon/3}$, for every seed node $s \in K$, if $|\mathrm{supp}(x^*)| \geq 2$ then we have that*

$$|\mathrm{supp}(x^*) \cap K^{\mathsf{c}}| > (1-\delta)\frac{p}{\gamma}(k-1)$$

*Proof.* If $|\mathrm{supp}(x^*)| \geq 2$ it means that $x_s^* > 0$ as otherwise we must have $x^* = 0$. Moreover, let $i \in V$ be such that $i \neq s$ and $x_i^* \geq x_j^*$ for all $j \neq s$. Then we must have that $i$ is a neighbor of $s$. Because $\Delta_i = 0$, $x_i^* > 0$ and $x_i^* \geq x_j^*$ for all $j \neq s$, we know that the amount of mass that node $s$ sends to node $i$ is strictly larger than 1, and hence $x_s^* > x_i^* + 1 > 1$. But then this means that we must have $x_\ell^* > 0$ for all $\ell \sim s$. By Lemma A.4 we know that with probability at least $1 - k^{-\epsilon/3}$, every node $i \in K$ has more than $(1-\delta)q(n-k)$ neighbors in $K^{\mathsf{c}}$. This applies to $s$ which was chosen arbitrarily from $K$. Therefore we have that with probability at least $1 - k^{-\epsilon/3}$, for every seed node $s \in K$, if $|\mathrm{supp}(x^*)| \geq 2$ then $|\mathrm{supp}(x^*) \cap K^{\mathsf{c}}| > (1-\delta)q(n-k)$. The required result then follows from our definition that $\gamma = \frac{p(k-1)}{q(n-k)}$. $\square$

On the other hand, Proposition A.3 provides an upper bound on the number of false positives incurred by $\mathrm{supp}(x^\dagger)$ under appropriately chosen source mass $\theta^\dagger$ at the seed node. Its proof is based on upper bounding the total amount of mass that leaks to the outside of the target cluster during a diffusion process, similar to the strategy used in the proof of Theorem 3.5 in Yang & Fountoulakis (2023).

**Proposition A.3.** *If $p \geq \max(\frac{(6+\epsilon_1)}{\delta_1^2} \frac{\log k}{k-2}, \frac{(\sqrt{8}+\epsilon_2)}{\delta_2\sqrt{1-\delta_1}} \frac{\sqrt{\log k}}{\sqrt{k-2}})$ for some $0 < \delta_1, \delta_2 \leq 1$ and $\epsilon_1, \epsilon_2 > 0$, then with probability at least $1 - 3k^{-\epsilon_1/6} - k^{-\epsilon_2}$, for every seed node $s \in K \cap \tilde{Y}_1$ with source mass*

$$\theta^\dagger = \left( \frac{a_1\gamma\frac{(k-2)}{(k-1)} + (1-a_0)}{a_1\gamma\frac{(k-2)}{(k-1)}} \right)^2 ra_1k,$$

*we have that $K \cap \tilde{Y}_1 \subseteq \mathrm{supp}(x^\dagger)$ and*

$$|\mathrm{supp}(x^\dagger) \cap K^{\mathsf{c}}| \leq \left( \left( \frac{a_1\gamma\frac{(k-2)}{(k-1)} + (1-a_0)}{a_1\gamma\frac{(k-2)}{(k-1)}} \right)^2 r - 1 \right) a_1k.$$

*Proof.* To see that $K \cap \tilde{Y}_1 \subseteq \mathrm{supp}(x^\dagger)$, let us assume for the sake of contradiction that $x_i^\dagger = 0$ for some $i \in K \cap \tilde{Y}_1$. This means that node $i$ receives at most 1 unit mass, because otherwise we would have $x_i^\dagger > 0$. We also know that $i \neq s$ because $\Delta_s > 1$. Denote $F := \{j \in K \cap \tilde{Y}_1 : j \sim s\}$. We will consider two cases depending on if $i \in F$ or not.

Suppose that $i \in F$. Then we have that $x_s^\dagger - x_i^\dagger \leq 1$ because node $i$ receives at most 1 unit mass from node $s$. This means that $x_s^\dagger \leq 1 + x_i^\dagger = 1$. It follows that the total amount of mass which flows out of node $s$ is

$$\sum_{\ell \sim s}(x_s^\dagger - x_\ell^\dagger) \leq \sum_{\ell \sim s} x_s^\dagger \leq \deg_{G^w}(s) \leq (1+\delta)(p(a_1k-1) + (1-a_0)q(n-k)),$$

where the last inequality follows from Lemma A.5. Therefore, we get that the total amount of source mass is at most

$$
\begin{aligned}
\theta^\dagger &\le (1+\delta)(p(a_1 k - 1) + (1 - a_0)q(n - k)) + 1 \\
&= (1+\delta)p(a_1 k - 1)\frac{a_1 p(k - 1/a_1) + (1 - a_0)q(n - k)}{a_1 p(k - 1/a_1)} + 1 \\
&= (1+\delta)p(a_1 k - 1)\frac{a_1\gamma^{\frac{(k - 1/a_1)}{(k-1)}} + (1 - a_0)}{a_1\gamma^{\frac{(k - 1/a_1)}{(k-1)}}} + 1 \\
&\le (1+\delta)p(a_1 k - 1)\frac{a_1\gamma^{\frac{(k-2)}{(k-1)}} + (1 - a_0)}{a_1\gamma^{\frac{(k-2)}{(k-1)}}} + 1 \\
&\le (1+\delta)(1 + 2/k)\left(\frac{a_1\gamma^{\frac{(k-2)}{(k-1)}} + (1 - a_0)}{a_1\gamma^{\frac{(k-2)}{(k-1)}}}\right) a_1 k < \theta^\dagger,
\end{aligned}
$$

where the second last inequality follows from $a_1 \ge 1/2$. This is a contradiction, and hence we must have $i \notin F$.

Now, suppose that $i \notin F$. Then we know that the total amount of mass that node $i$ receives from its neighbors is at most 1. In particular, node $i$ receives at most 1 unit mass from nodes in $F$. This means that

$$
\sum_{\substack{j \sim i \\ j \in F}} x_j^\dagger = \sum_{\substack{j \sim i \\ j \in F}} (x_j^\dagger - x_i^\dagger) \le 1.
$$

By Lemma A.6, we know that with probability at least $1 - 2k^{-\epsilon_1/6} - k^{-\epsilon_2}$, node $i$ has at least $(1 - \delta_1)(1 - \delta_2)p^2(a_1 k - 1)$ neighbors in $F$, and thus

$$
\sum_{\substack{j \in F \\ j \sim i}} x_j^\dagger \le 1 \implies \min_{j \in F} x_j^\dagger \le \frac{1}{(1 - \delta_1)(1 - \delta_2)p^2(a_1 k - 1)}
$$

Therefore, let $j \in F$ a node such that $x_j^\dagger \le x_\ell^\dagger$ for all $\ell \in F$, then with probability at least $1 - 2k^{-\epsilon_1/6} - k^{-\epsilon_2}$,

$$
x_j^\dagger \le \frac{1}{(1 - \delta_1)(1 - \delta_2)p^2(a_1 k - 1)}. \tag{10}
$$

By Lemma A.6, with probability at least $1 - 2k^{-\epsilon_1/6} - k^{-\epsilon_2}$, node $j$ has at least $(1 - \delta_1)(1 - \delta_2)p^2(a_1 k - 1) - 1$ neighbors in $F$. Since $x_j^\dagger \le x_\ell^\dagger$ for all $\ell \in F$ and $x_j^\dagger \le x_s^\dagger$, we know that

$$
|\{\ell \in V : \ell \sim j \text{ and } x_\ell^\dagger \ge x_j^\dagger\}| \ge (1 - \delta_1)(1 - \delta_2)p^2(a_1 k - 1). \tag{11}
$$

Therefore, with probability at least $1 - 3k^{-\epsilon_1/3} - k^{-\epsilon_2}$, the total amount of mass that node $j$ sends out to its neighbors is at most

$$
\begin{aligned}
\sum_{\ell \sim j} (x_j^\dagger - x_\ell^\dagger) &\le \sum_{\substack{\ell \sim j \\ x_\ell^\dagger \le x_j^\dagger}} (x_j^\dagger - x_\ell^\dagger) \le \sum_{\substack{\ell \sim j \\ x_\ell^\dagger \le x_j^\dagger}} x_j^\dagger \\
&\overset{(i)}{\le} \left((1 + \delta_1)(p(a_1 k - 1) + (1 - a_0)q(n - k)) - (1 - \delta_1)(1 - \delta_2)p^2(a_1 k - 1)\right)x_j^\dagger \\
&\overset{(ii)}{\le} \frac{(1 + \delta_1)}{(1 - \delta_1)(1 - \delta_2)}\left(\frac{p(a_1 k - 1) + (1 - a_0)q(n - k)}{p^2(a_1 k - 1)}\right) - 1.
\end{aligned}
$$

where (i) follows from Lemma A.5 and (11), and (ii) follows from (10). Since node $j$ settles 1 unit mass, the total amount of mass that node $j$ receives from its neighbors is therefore at most

$$
\frac{(1 + \delta_1)}{(1 - \delta_1)(1 - \delta_2)}\left(\frac{p(a_1 k - 1) + (1 - a_0)q(n - k)}{p^2(a_1 k - 1)}\right).
$$

Recall that the amount of mass that node $j$ receives from node $s$ is given by $x_s^\dagger - x_j^\dagger$, and hence we get

$$x_s^\dagger \leq \frac{(1+\delta_1)}{(1-\delta_1)(1-\delta_2)} \left( \frac{p(a_1 k - 1) + (1-a_0)q(n-k)}{p^2(a_1 k - 1)} \right) + x_j^\dagger. \tag{12}$$

Apply the same reasoning as before, we get that with probability at least $1 - 3k^{-\epsilon_1/6} - k^{-\epsilon_2}$, the total amount of mass that is sent out from node $s$ is

$$\sum_{\ell \sim s}(x_s^\dagger - x_\ell^\dagger) \;<\; \deg_{G^w}(s) \cdot x_s^\dagger \overset{(i)}{\leq} (1+\delta_1)(p(a_1 k - 1) + (1-a_0)q(n-k)) \cdot x_s^\dagger$$

$$\overset{(ii)}{\leq} \frac{(1+\delta_1)}{(1-\delta_1)(1-\delta_2)} \left( (1+\delta_1)\frac{(p(a_1 k - 1) + (1-a_0)q(n-k))^2}{p^2(a_1 k - 1)^2} \right.$$
$$\left. + \frac{p(a_1 k - 1) + (1-a_0)q(n-k)}{p^2(a_1 k - 1)^2} \right)(a_1 k - 1)$$

$$\overset{(iii)}{\leq} \frac{(1+\delta_1)(1+\delta_1 + \frac{2}{p(k-1)})}{(1-\delta_1)(1-\delta_2)} \left( \frac{p(a_1 k - 1) + (1-a_0)q(n-k)}{p(a_1 k - 1)} \right)^2 (a_1 k - 1)$$

$$\overset{(iv)}{=} \frac{(1+\delta_1)(1+\delta_1 + \frac{2}{p(k-1)})}{(1-\delta_1)(1-\delta_2)} \left( \frac{a_1\gamma\frac{(k-1/a_1)}{(k-1)} + (1-a_0)}{a_1\gamma\frac{(k-1/a_1)}{(k-1)}} \right)^2 (a_1 k - 1)$$

$$\overset{(v)}{\leq} \frac{(1+\delta_1)(1+\delta_1 + \frac{2}{p(k-1)})}{(1-\delta_1)(1-\delta_2)} \left( \frac{a_1\gamma\frac{(k-2)}{(k-1)} + (1-a_0)}{a_1\gamma\frac{(k-2)}{(k-1)}} \right)^2 (a_1 k - 1),$$

where (i) follows from Lemma A.5, (ii) follows from (10) and (12), (iii) and (v) uses $a_1 \geq 1/2$, and (iv) follows from the definition $\gamma = \frac{p(k-1)}{q(n-k)}$. This implies that the total amount of source mass is

$$\theta^\dagger < \frac{(1+\delta_1)(1+\delta_1 + \frac{2}{p(k-1)})}{(1-\delta_1)(1-\delta_2)} \left( \frac{a_1\gamma\frac{(k-2)}{(k-1)} + (1-a_0)}{a_1\gamma\frac{(k-2)}{(k-1)}} \right)^2 a_1 k = \theta^\dagger$$

which is a contradiction. Therefore we must have $i \notin K \cap \tilde{Y}_1$, but then this contradicts our assumption that $i \in K \cap \tilde{Y}_1$. Since our choice of $i, s \in K_1$ were arbitrary, this means that $x_i^\dagger > 0$ for all $i \in K_1$ and for all $s \in K_1$.

Finally, the upper bound on $|\mathrm{supp}(x^\dagger) \cap K^c|$ follows directly from the fact that $x_i^\dagger > 0$ only if node $i$ settles 1 unit mass. $\qquad\square$

By Proposition A.2, the F1 score for $\mathrm{supp}(x^*)$ is at most

$$\mathrm{F1}(\mathrm{supp}(x^*)) < \frac{2k}{2k + (1-\delta_3)p(k-1)/\gamma}.$$

By Proposition A.3, the F1 score for $\mathrm{supp}(x^\dagger)$ is at least

$$\mathrm{F1}(\mathrm{supp}(x^\dagger)) \geq \frac{2k}{2k + \left( \left( \frac{a_1\gamma\frac{(k-2)}{(k-1)} + (1-a_0)}{a_1\gamma\frac{(k-2)}{(k-1)}} \right)^2 r - 1 \right) a_1 k + (1-a_1)k}.$$

Therefore, a sufficient condition for $\text{F1}(\text{supp}(x^\dagger)) \geq \text{F1}(\text{supp}(x^*))$ is

$$\left( \left( \frac{a_1 \gamma \frac{(k-2)}{(k-1)} + (1-a_0)}{a_1 \gamma \frac{(k-2)}{(k-1)}} \right)^2 r - 1 \right) a_1 + (1-a_1) \leq (1-\delta_3)\frac{p}{\gamma}\frac{(k-1)}{k}$$

$$\iff \left( \frac{a_1 \gamma \frac{(k-2)}{(k-1)} + (1-a_0)}{\gamma \frac{(k-2)}{(k-1)}} \right)^2 \leq \frac{a_1}{r}\left( (1-\delta_3)\frac{p}{\gamma}\frac{(k-1)}{k} + 2a_1 - 1 \right)$$

$$\iff a_1 \gamma \frac{(k-2)}{(k-1)} + 1 - a_0 \leq \gamma \frac{(k-2)}{(k-1)}\sqrt{\frac{a_1}{r}\left( (1-\delta_3)\frac{p}{\gamma}\frac{(k-1)}{k} + 2a_1 - 1 \right)}$$

$$= \gamma \frac{(k-2)}{(k-1)}\sqrt{\left( \frac{p/\gamma}{r'} + \frac{2a_1-1}{r} \right)a_1}$$

$$\iff a_0 \geq 1 - \frac{(k-2)}{(k-1)}\left( \sqrt{\left( \frac{p/\gamma}{r'} + \frac{2a_1-1}{r} \right)a_1} - a_1 \right)\gamma.$$

Finally, setting $K' = K \cap \tilde{Y}_1$ completes the proof of Theorem A.1.

## A.1 CONCENTRATION RESULTS

**Lemma A.4** (External degree in $G$). *If $q \geq \frac{(3+\epsilon)}{\delta^2}\frac{\log k}{n-k}$ for some $\epsilon > 0$ and $0 < \delta \leq 1$, then with probability at least $1 - k^{-\epsilon/3}$ we have that for all $i \in K$,*

$$|E(\{i\}, K^c)| \geq (1-\delta)q(n-k).$$

*Proof.* This follows directly by noting that, for each $i \in K$, $|E(\{i\}, K^c)|$ is the sum of independent Bernoulli random variables with mean $q(n-k)$. Applying a multiplicative Chernoff bound on $|E(\{i\}, K^c)|$ and then a union bound over $i \in K$ gives the result. $\square$

**Lemma A.5** (Node degree in $G^w$). *If $p \geq \frac{(6+\epsilon)}{\delta^2}\frac{\log k}{k-2}$ for some $\epsilon > 0$ and $0 < \delta \leq 1$, then with probability at least $1 - k^{-\epsilon/6}$ we have that for all $i \in K \cap \tilde{Y}_1$,*

$$\deg_{G^w}(i) \leq (1+\delta)(p(a_1 k - 1) + (1-a_0)q(n-k)).$$

*Proof.* For each node $i \in K \cap \tilde{Y}_1$, since $K \cap \tilde{Y}_1 = a_1 k$ and $K^c \cap \tilde{Y}_1 = (1-a_0)(n-k)$, its degree in $G^w$, that is $\deg_{G^w}(i)$, is the sum of independent Bernoulli random variables with mean $\mathbb{E}(\deg_{G^w}(i)) = p(a_1 k - 1) + (1-a_0)q(n-k) \geq p(a_1 k - 1) \geq \frac{(3+\epsilon/2)}{\delta^2}\log k$. Apply the Chernoff bound we get

$$\mathbb{P}\left( \deg_{G^w}(i) \geq (1+\delta)\mathbb{E}(\deg_{G^w}(i)) \right) \leq \exp(-\delta^2 \mathbb{E}(\deg_{G^w}(i))/3) \leq \exp(-(1+\epsilon/6)\log k).$$

Taking a union bound over all $i \in K \cap \tilde{Y}_1$ gives the result. $\square$

**Lemma A.6** (Internal connectivity in $G^w$). *If $p \geq \max(\frac{(6+\epsilon_1)}{\delta_1^2}\frac{\log k}{k-2}, \frac{(\sqrt{8}+\epsilon_2)}{\delta_2\sqrt{1-\delta_1}}\frac{\sqrt{\log k}}{\sqrt{k-2}})$, then with probability at least $1 - 2k^{-\epsilon_1/6} - k^{-\epsilon_2}$, we have that for all $i, j \in K \cap \tilde{Y}_1$ where $i \neq j$, there are at least $(1-\delta_1)(1-\delta_2)p^2(a_1 k - 1)$ distinct paths connecting node $i$ to node $j$ such that, each of these paths consists of at most 2 edges, and each edge from $G^w$ appears in at most one of these paths.*

*Proof.* Let $F_i$ denote the set of neighbors of a node $i$ in $K \cap \tilde{Y}_1$. By our assumption that $p \geq \frac{(6+\epsilon_1)}{\delta_1^2}\frac{\log k}{k-2}$, we may take a Chernoff bound on the size of $F_i$ and a union bound over all $i \in K \cap \tilde{Y}_1$ to get that, with probability at least $1 - 2k^{-\epsilon_1/6}$,

$$(1-\delta_1)p(a_1 k - 1) \leq |F_i| \leq (1+\delta_1)p(a_1 k - 1), \; \forall i \in K \cap \tilde{Y}_1.$$

If $j \notin F_i$, then since $|E(\{j\}, F_i)|$ is a sum of independent Bernoulli random variables with mean $|F_i|p$, we may apply the Chernoff bound and get that, with probability at least $1 - 2k^{-\epsilon_1/6}$ (under the event that $(1 - \delta_1)p(a_1 k - 1) \leq |F_i| \leq (1 + \delta_1)p(a_1 k - 1)$),

$$\mathbb{P}(|E(\{j\}, F_i)| \leq (1 - \delta_2)|F_i|p) \leq \exp(-\delta_2^2 |F_i|p/2) \leq \exp(-\delta_2^2 (1 - \delta_1)p^2 (a_1 k - 1)/2))$$
$$\leq \exp(-(2 + \epsilon_2) \log k). \quad (13)$$

The last inequality in the above follows from our assumption that $p \geq \frac{(\sqrt{8} + \epsilon_2)}{\delta_2 \sqrt{1 - \delta_1}} \frac{\sqrt{\log k}}{\sqrt{k - 2}}$. If $j \in F_i$, then the edge $(i, j)$ is a path of length 1 connecting $j$ to $i$, and moreover, let $\ell \in K \cap \tilde{Y}_1$ be such that $\ell \notin F_i$ and $\ell \neq i$, we have that

$$\mathbb{P}(|E(\{j\}, F_i \backslash \{j\})| + 1 \leq (1 - \delta_2)|F_i|p) \leq \mathbb{P}(|E(\{\ell\}, F_i)| \leq (1 - \delta_2)|F_i|p)$$
$$\leq \exp(-(2 + \epsilon_2) \log k),$$

where the last inequality follows from (13). Note that, for a node $j$ in $K \cap \tilde{Y}_1$ such that $j \neq i$, each edge $(j, \ell) \in E(\{j\}, F_i \backslash \{j\})$ identifies a unique path $(j, \ell, i)$ and none of these paths has overlapping edges. Therefore, denote $P(i, j)$ the set of mutually non-overlapping paths of length at most 2 between $i$ and $j$, and take union bound over all $i, j \in K \cap \tilde{Y}_1$, we get that

$$\mathbb{P}(\exists i, j \in K \cap \tilde{Y}_1, i \neq j, \text{s.t. } P(i, j) \leq (1 - \delta_2)|F_i|p) \leq k^{-\epsilon_2}.$$

Finally, taking a uninon bound over the above event and the event that there is $i \in K \cap \tilde{Y}_1$ such that $|F_i| < (1 - \delta_1)p(a_1 k - 1)$ gives the required result. $\qquad \square$

# B   Discussion: How to set edge weight $\epsilon$ in $G^w$ under the local random model (Definition 3.2)

Given a graph $G = (V, E)$ generated from the local random model described in Definition 3.2, noisy labels $\tilde{y}_i$ for node $i \in V$, recall from (4) that the edge weights in $G^w$ are such that $w((i, j)) = 1$ if $\tilde{y}_i = \tilde{y}_j$ and $w((i, j)) = \epsilon$ otherwise. Our analysis in Section 3 takes $\epsilon = 0$. While understanding diffusion in $G^w$ with $\epsilon = 0$ already provides us with some insights with regard to how noisy labels can be useful, a natural extension of our analysis is to determine an "optimal" $\epsilon$ given model parameters $n, k, p, q$ and label accuracy $a_0, a_1$.

To see why this is an interesting problem, consider the case when $a_1$ is low. In this case, if we set $\epsilon = 0$ and start diffusing mass from a seed node $s \in K$ with $\tilde{y}_s = 1$, then diffusion in $G^w$ cannot reach a node $i \in K$ such that $\tilde{y}_i = 0$, because the graph $G^w$ is disconnected with two components $\tilde{Y}_1 = \{i \in V : \tilde{y}_i = 1\}$ and $\tilde{Y}_0 = \{i \in V : \tilde{y}_i = 0\}$. Consequently, the recall of the output cluster is at most $a_1$. On the other hand, if we instead set $\epsilon > 0$, this allows diffusion in $G^w$ to reach node $i \in K$ whose label is $\tilde{y}_i = 0$, however, at the same time, diffusion in $G^w$ incurs the risk of reaching a node $i \notin K$ whose label is $\tilde{y}_i = 0$. Therefore, setting $\epsilon > 0$ allows for discovering more true positives at the expense of incurring more false positives. Whether one should set $\epsilon = 0$ will depend on $n, k, p, q, a_0, a_1$ and the accuracy metric (e.g. precision, recall, or the F1) one aims to maximize. If the objective is to maximize recall, then it is easy to see that one should set $\epsilon > 0$ to allow recovering nodes in $K$ that receive different noisy labels. In general, it turns out that rigorously characterizing an "optimal" $\epsilon$ that maximizes other accuracy metrics such as the F1 is nontrivial. In what follows we discuss intuitively the potential conditions under which one should set $\epsilon = 0$ or $\epsilon > 0$ to obtain a better clustering result which balances precision and recall, i.e. attains a higher F1. In addition, we empirically demonstrate these conditions over synthetic data.

***Conjecture 1:*** *A sufficient condition to favor $\epsilon > 0$ is $(1 - a_1)pk > a_0 q(n - k)$.*

***Conjecture 2:*** *A sufficient condition to favor $\epsilon = 0$ is $(1 - a_1)p^2 k < a_0 q^2(n - k)$.*

We provide an informal explanation for these conditions. Note that if a seed node $s$ is drawn uniformly at random from $K$, then with probability $a_1 \geq 1/2$ we get that $s \in K \cap \tilde{Y}_1$. Therefore let us assume that a seed node is selected from $K \cap \tilde{Y}_1$. In this case, since the diffusion of mass starts from within $K \cap \tilde{Y}_1$, excess mass needs to get out of $K \cap \tilde{Y}_1$ along the cut edges of $K \cap \tilde{Y}_1$. Let us focus on the edges between $K \cap \tilde{Y}_1$ and $\tilde{Y}_0$ since these are the edges affected by $\epsilon$. The edges between

$K \cap \tilde{Y}_1$ and $K^{\mathsf{c}} \cap \tilde{Y}_1$ are not affected by $\epsilon$. In expectation, every node in $K \cap \tilde{Y}_1$ has $(1-a_1)pk$ number of neighbors in $K \cap \tilde{Y}_0$ and $a_0 q(n-k)$ number of neighbors in $K^{\mathsf{c}} \cap \tilde{Y}_0$. Therefore, in a diffusion step when we push excess mass from a node in $K \cap \tilde{Y}_1$ to its neighbors, for every $(1-a_1)pk$ unit mass that is pushed into $K \cap \tilde{Y}_0$, on average $a_0 q(n-k)$ unit mass is pushed into $K^{\mathsf{c}} \cap \tilde{Y}_0$. If $(1-a_1)pk > a_0 q(n-k)$, then this means that $K \cap \tilde{Y}_0$ receives more mass than $K^{\mathsf{c}} \cap \tilde{Y}_0$. Consequently, as a result of $K \cap \tilde{Y}_0$ receiving more mass than $K^{\mathsf{c}} \cap \tilde{Y}_0$ from a diffusion step within $K \cap \tilde{Y}_1$, we can expect that by setting $\epsilon > 0$, we obtain more true positives as the diffusion process covers nodes in $K \cap \tilde{Y}_0$ at the expense of fewer false positives as the diffusion process covers less nodes in $K^{\mathsf{c}} \cap \tilde{Y}_0$. This leads to our first conjecture on the condition to favor $\epsilon > 0$ over $\epsilon = 0$.

For the condition in our second conjecture, note that even when $K \cap \tilde{Y}_0$ receives less mass from $K \cap \tilde{Y}_1$ than the amount of mass that $K^{\mathsf{c}} \cap \tilde{Y}_0$ receives from $K \cap \tilde{Y}_1$, it does not necessarily imply that we would get more number of false negatives from $K^{\mathsf{c}} \cap \tilde{Y}_0$ and fewer number of true positives from $K \cap \tilde{Y}_0$. Recall that we use $\mathrm{supp}(x^*)$ as the output cluster where $x^*$ is the optimal solution of the diffusion problem (2). For a node $i \in V$, we know that $x_i^* > 0$ only if node $i$ receives more than 1 unit mass. Consider the following two average diffusion dynamics. First, as discussed before, for every $(1-a_1)pk$ unit mass that is pushed into $K \cap \tilde{Y}_0$ from the $a_1 pk$ nodes in $K \cap \tilde{Y}_1$, on average (i.e. averaged over multiple nodes) $a_0 q(n-k)$ unit mass is pushed into $K^{\mathsf{c}} \cap \tilde{Y}_0$. Second, in expectation, every node in $K \cap \tilde{Y}_0$ has $a_1 pk$ neighbors in $K \cap \tilde{Y}_1$ and every node in $K^{\mathsf{c}} \cap \tilde{Y}_0$ has $a_1 qk$ neighbors in $K \cap \tilde{Y}_1$. For every unit mass that moves from $K \cap \tilde{Y}_1$ to $K \cap \tilde{Y}_0$, a node in $K \cap \tilde{Y}_0$ on average (i.e. averaged over multiple nodes and multiple diffusion steps) receives $pa_1 k/a_1 k = p$ unit and a node in $K^{\mathsf{c}} \cap \tilde{Y}_0$ on average receives $qa_1 k/a_1 k = q$ unit. Combining the above two points, we get that on average, for every $(1-a_1)p^2 k$ unit mass received by a node in $K \cap \tilde{Y}_0$, a node in $K^{\mathsf{c}} \cap \tilde{Y}_0$ receives $a_0 q^2 (n-k)$ unit mass. Therefore, if $(1-a_1)p^2 k < a_0 q^2 (n-k)$, then setting $\epsilon > 0$ would make a node $i \in K^{\mathsf{c}} \cap \tilde{Y}_0$ generally receive less mass than a node in $j \in K^{\mathsf{c}} \cap \tilde{Y}_0$. Consequently, a node $j \in K^{\mathsf{c}} \cap \tilde{Y}_0$ is more likely to receive more than 1 unit mass. This implies that, by setting $\epsilon > 0$ we would get fewer number of true positives from $K \cap \tilde{Y}_0$ at the expense of incurring more number of false positives from $K^{\mathsf{c}} \cap \tilde{Y}_0$. This leads to our second conjecture.

Of course, a rigorous argument to justify both conjectures will require a much more careful analysis of the diffusion dynamics and additional assumptions on $p, q$ so that the average behaviors described in the above hold with high probability. In addition, there is a gap of order $p/q$ between the two conditions. It is also an interesting question to determine a good strategy to set $\epsilon$ in that "gap regime". Addressing these questions are nontrivial and we leave it for future work.

### B.1 EMPIRICAL DEMONSTRATION OF OUR CONJECTURES

We demonstrate our conjectures on when to set $\epsilon = 0$ or $\epsilon > 0$ over synthetic data. As in Section 4.1, we generate synthetic graphs using the stochastic block model with cluster size $k = 500$ an number of clusters $c = 20$. The number of nodes in the graph equals $n = kc = 10,000$. Two nodes within the same cluster are connected with probability $p$ and two nodes from different clusters are connected with probability $q$. We consider different choices for $q, a_0, a1$ such that the condition in either Conjecture 1 or Conjecture 2 is satisfied. Other empirical settings are the same as in Section 4.1.

In Table 2 we report the F1 scores obtained by setting $\epsilon = 0$ and $\epsilon = 0.2$, respectively. We average over 100 trials for each setting. For comparison purposes we also include the results obtained by employing Flow Diffusion (FD) over the original graph. Note that FD is equivalent to setting $\epsilon = 1$. Observe that, when $(1-a_1)pk > a_0 q(n-k)$ as required by Conjecture 1, setting $\epsilon = 0.2$ leads to a higher F1, whereas when $(1-a_1)p^2 k < a_0 q^2 (n-k)$ as required by Conjecture 2, setting $\epsilon = 0$ leads to a higher F1. This demonstrates both conjectures.

From a practical point of view, we would like to remark that real networks often have much more complex structures than the synthetic graphs. Therefore the same conditions may not generalize to the real networks that one would work with in practice. To that end, in Section C.1 we provide an empirical study on the robustness of our method with respect to different values of $\epsilon$. Our empirical results in Section C.1 indicate that, over real networks, the local clustering accuracy remains similar for different choices of $\epsilon$, ranging from 0.01 to 0.2.

Table 2: Empirical demonstration of our conjectures: F1 scores for local clustering in the local random graph model with different model parameters and label accuracy

|  | Empirical Setting | FD | LFD ($\epsilon$=0) | LFD ($\epsilon$=0.2) |
|---|---|---|---|---|
| Conjecture 1 | $p$=0.05, $q$=0.0015, $a_0$=0.7, $a_1$=0.6 | 69.2 | 64.5 | **77.8** |
|  | $p$=0.05, $q$=0.0015, $a_0$=0.6, $a_1$=0.65 | 69.2 | 64.2 | **74.6** |
| Conjecture 2 | $p$=0.05, $q$=0.0075, $a_0$=0.8, $a_1$=0.7 | 9.7 | **48.8** | 37.3 |
|  | $p$=0.05, $q$=0.0075, $a_0$=0.9, $a_1$=0.9 | 9.7 | **76.7** | 61.1 |

## C   FURTHER EVALUATIONS AND COMPARISONS

### C.1   HYPERPARAMETER ANALYSIS

In this section, we test the robustness of our method against various choices of hyperparameters. There are 2 hyperparameters in our method. The first hyperparameter is the edge weight $\epsilon \in [0, 1)$ from (4), and the second hyperparameter is the total amount of source mass $\theta > 0$ at the seed node(s) to initialize the flow diffusion process.

We conduct a detailed case study using the Coauthor CS dataset. Similar trends and results are seen in the experiments using other datasets. We focus on the one of the empirical settings considered in Section 4.2, where we are given 10 positive and 10 negative ground-truth node labels. Apart from the choices for $\epsilon$ and $\theta$, we keep all other empirical settings the same as in Section 4.2. We report the average local clustering result over 100 trails.

We vary the total amount of source mass $\theta$ as follows. Set $\theta = \alpha \text{vol}_G(K)$ and we let $\alpha \in \{2, 3, 4, 5\}$. Recall that $K$ denotes the target cluster. Therefore picking $\alpha$ in the range of $[2, 5]$ results in very large variations in $\theta$. The experiments in Section 4.2 use $\alpha = 2$ and $\epsilon = 0.05$. Here, we present results using different combinations of values for $\alpha$ and $\epsilon$. Observe that for a fixed $\alpha \in \{2, 3, 4\}$, the maximum change in the F1 score across $\epsilon \in [10^{-2}, 10^{-1}]$ is 1.2. Moreover, for all combinations of $\epsilon$ and $\alpha$, our method has a much higher F1, highlighting the effectivess and robustness to incorporate noisy labels for local clustering.

Table 3: F1 scores for different values of source mass and inter-edge weight

| $\alpha$ | FD | WFD | LFD | | | | | |
|---|---|---|---|---|---|---|---|---|
|  |  |  | $\epsilon = 0.01$ | $\epsilon = 0.025$ | $\epsilon = 0.05$ | $\epsilon = 0.075$ | $\epsilon = 0.1$ | $\epsilon = 0.2$ |
| 2 | 62.8 | 56.4 | 73.0 | **73.1** | 72.6 | 72.4 | 72.2 | 70.8 |
| 3 | 67.5 | 58.3 | **74.8** | 74.1 | 74.1 | 74.0 | 74.0 | 73.0 |
| 4 | 68.1 | 57.6 | **73.4** | 72.7 | 72.1 | 72.3 | 72.2 | 72.0 |
| 5 | 66.1 | 55.4 | **71.8** | 70.8 | 70.0 | 69.5 | 69.3 | 68.7 |

### C.2   RUNTIME ANALYSIS

We report the running time of our Label-based Flow Diffusion (LFD) along with other flow diffusion-based local methods. The experiments are run on an Intel i9-13900K CPU with 36MB Cache and 2 x 48GB DDR5 RAM. We highlight the fast running time of LFD. Fast running times are typically seen in local methods and are due to the fact that these methods do not require processing the entire graph. The runtimes reported in Table 4 are based on the experiments using the Coauthor CS dataset, averaged over 10 trials across 15 clusters.

### C.3   COMPARISON WITH GRAPH CONVOLUTIONAL NETWORKS (GCNS)

Within the task of clustering or node classification in the presence of ground-truth node labels, it is natural to extend the comparison to other types of methods which exploit both the graph structure and node information. To that end, we provide an empirical comparison with the performance of GCNs in our problem setting. Note that both the training and the inference stages of GCNs require

Table 4: Average runtimes for different local diffusion methods

|  | FD(Fountoulakis et al., 2020) | WFD (Yang & Fountoulakis, 2023) | LFD (ours) |
|---|---|---|---|
| Train model | - | - | $0.09 \pm 0.01$ s |
| Calculate weights | - | $1.11 \pm 0.77$ s | $0.03 \pm 0.02$ s |
| Diffusion process | $0.01 \pm 0.01$ s | $0.01 \pm 0.01$ s | $0.01 \pm 0.01$ s |
| TOTAL | $0.01 \pm 0.01$ s | $1.12 \pm 0.78$ s | $0.13 \pm 0.03$ s |

accessing every node in the graph, and this makes GCNs (and more generally other global methods that require full graph processing) unsuitable for local graph clustering. In contrast, local methods only explore a small portion of the graph around the seed node(s).

To highlight the strengths of our local method against global methods such as GCNs, we carried out additional experiments to compare both runtime and accuracy. Again, we fix the same empirical setting as before, that is, we use the Coauthor CS dataset and select 10 nodes each from positive and negative ground-truth categories. Let us remind the reader that, here, a positive ground-truth label means that a node selected from the target cluster $K$ is given a label 1. Similarly, a negative ground-truth label means that a node selected from the rest of the graph is given a label 0.

Table 5: Comparison between Label-based Flow Diffusion (LFD) and Graph Convolutional Network (GCN)

|  | LFD | GCN |
|---|---|---|
| F1-score | 73.0 | 46.9 |
| Runtime | $0.13 \pm 0.03$ s | $3.68 \pm 0.31$ s |

We use a two-layer GCN architecture with a hidden layer size of 16. When training the GCN model, we terminate the training process after 100 epochs. In our approach, each class is treated separately in a one-vs-all classification framework during training. We replicate this procedure for each class across 10 independent trials. The results are shown in Table 5. Observe that our method not only runs substantially faster (i.e. 28 times faster) than a GCN but also obtains a much higher F1. The poor performance of GCNs is due to the scarcity of ground-truth data in our setting, where we only have 20 samples. GCNs generally require a much greater number of ground-truth labels to work well. In order to make GCN achieve a better accuracy than LFD, we had to increase the number of ground-truth labels to 600 samples, which is not very realistic for local clustering contexts.

## D EXPERIMENTS

### D.1 REAL-WORLD DATASET DESCRIPTION

- Coauthor CS is a co-authorship graph based on the Microsoft Academic Graph from the KDD Cup 2016 challenge (Shchur et al. (2018)). Each node in the graph represents an author, while an edge represents the co-authorship of a paper between two authors. The ground-truth node labels are determined by the most active research field of each author. The Coauthor CS graph consists of 18,333 computer science authors with 81,894 connections and 15 ground-truth clusters.

- Coauthor Physics is a co-authorship graph also extracted from the Microsoft Academic Graph and used in the KDD Cup 2016 challenge (Shchur et al. (2018)). Its structure is similar to Coauthor CS with a focus on Physics research. The dataset has 34,493 physics researchers and 247,962 connections among them. Each physics researcher belongs to one of the 5 ground-truth clusters.

- Amazon Photo is a co-purchasing graph from Amazon (McAuley et al. (2015)), where nodes represent products and an edge indicates whether two products are frequently bought together. Labels of the nodes are determined by the product's category, while node attributes are bag-of-word encodings of product reviews. The dataset consists of 7,487 photographic equipment products, 119,043 co-purchasing connections, and 8 categories

- Amazon Computers is another co-purchasing graph extracted from Amazon (Shchur et al. (2018)), with the same structure as Amazon Photo. It has 13,752 computer equipment products, 245,861 connections, and 10 categories.

- Cora (McCallum et al. (2000)) is a citation network where each node denotes a scientific publication in Computer Science. An edge from node A to B indicates a citation from work A to work B. Despite their directed nature, we utilize an undirected version of these graphs for our analysis. The graph includes 2,708 publications, 5,429 edges, and 7 classes denoting the paper categories. The node features are bag-of-words encodings of the paper abstract.

- Pubmed (Sen et al. (2008)) is a citation network with a similar structure as Cora. We also adopt an undirected version of the graph. The dataset categorizes medical publications into one of 3 classes and comprises 19,717 nodes and 44,338 edges. Node features are TF/IDF encodings from a selected dictionary.

### D.2 Beyond flow diffusion: empirical validations with PageRank

In this section, we extend the comparisons beyond just flow diffusion, considering another local graph clustering technique, namely PageRank. We employ the $\ell_1$-regularized PageRank (Fountoulakis et al., 2017), demonstrating that the outcomes align consistently with those of flow diffusion. In the next sections, findings are reported for both Flow Diffusion (FD) and PageRank (PR).

Synthetic experiments

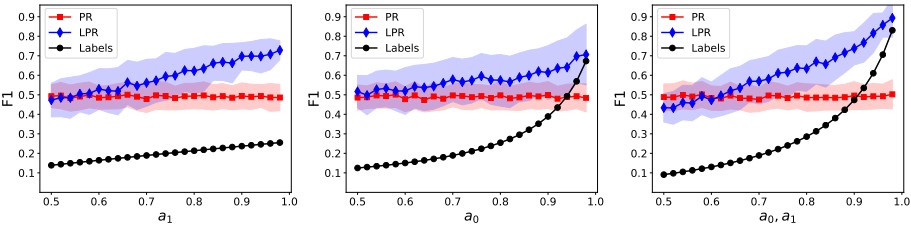

Figure 4: F1 scores obtained by employing $\ell_1$-regularized PageRank (Fountoulakis et al. (2017)) over the original graph (PR) and the label-weighted graph (LPR). For comparison, we also plot the F1 obtained by the noisy labels (Labels). The solid line and error bar show mean and standard deviation over 100 trials, respectively.

Real-world experiments: Label-based PageRank with ground-truth data

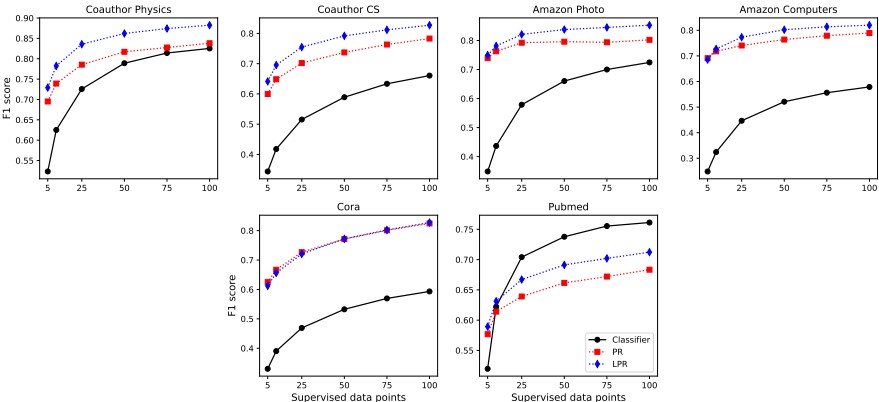

Figure 5: F1 scores across datasets for $\ell_1$-regularized PageRank, Label-based PageRank (LPR), and Logistic Regression (Classifier) with an increasing number of positive and negative ground truth samples

Real-world experiments: single seed Label-based PageRank with no ground-truth labels

Table 6: Comparison of F1 scores across datasets for PageRank (PR) and Label-based PageRank (LPR) in the absence of supervised data.

| Dataset | PR (single-seed) | PR (multi-seed) | LPR | Improv. ($\pm$) | Improv. (%) |
|---|---|---|---|---|---|
| Coauthor CS | 52.8 | 56.4 | **66.2** | +9.8 | +17.3 |
| Coauthor Physics | 63.4 | 61.9 | **72.1** | +8.7 | +13.6 |
| Amazon Photo | 64.2 | 63.8 | **67.6** | +3.4 | +5.3 |
| Amazon Computers | 57.7 | 60.4 | **63.2** | +2.8 | +4.7 |
| Cora | 55.7 | 58.5 | **60.6** | +2.1 | +3.5 |
| Pubmed | 56.1 | 54.9 | **58.8** | +2.7 | +4.8 |
| AVERAGE | 58.3 | 59.3 | 64.7 | +4.9 | +8.2 |

### D.3 DETAILED RESULTS OF EXPERIMENT WITH GROUND-TRUTH DATA

In this subsection, we present detailed results from the first experiment with real-world data. We report the performance of the setting with 25 positive and 25 negative nodes. We employ a Logistic Regression model with $\ell_2$ regularization for binary classification. During inference, labels form a weighted graph as described in 4, with $\epsilon = 0.05$, applied only over existing edges. In flow diffusion, the source mass of each seed node is assigned to be twice the volume of the target cluster. For PageRank, the starting scores of the source nodes are proportional to their degrees. The $\ell_1$-regularization parameter for PageRank is set to be the inverse of the total mass dispersed in flow diffusion. Additionally, we execute a line search process to determine the optimal teleportation parameter for PageRank. After finishing each diffusion process, a sweep-cut procedure is conducted on the resulting embeddings using the unweighted graph.

Table 7: F1 scores for the Coauthor CS dataset with 25 positive and 25 negative nodes.

| | Cluster | CLF | FD | WFD | LFD | PR | LPR |
|---|---|---|---|---|---|---|---|
| 1 | Bioinformatics | 85.5 | 45.8 | 55.6 | 61.9 | 51.6 | 65.5 |
| 2 | Machine Learning | 26.8 | 50.2 | 49.6 | 63.9 | 54.1 | 61.3 |
| 3 | Computer Vision | 79.4 | 64.8 | 38.5 | 82.4 | 60.9 | 71.2 |
| 4 | NLP | 16.7 | 58.2 | 73.5 | 73.0 | 68.6 | 76.6 |
| 5 | Graphics | 52.8 | 76.8 | 75.5 | 85.7 | 74.1 | 79.2 |
| 6 | Networks | 79.5 | 67.7 | 64.4 | 80.7 | 64.2 | 73.3 |
| 7 | Security | 38.3 | 49.4 | 58.5 | 62.5 | 57.3 | 62.7 |
| 8 | Databases | 39.6 | 73.0 | 72.0 | 81.0 | 75.2 | 75.1 |
| 9 | Data mining | 49.4 | 43.4 | 42.6 | 63.0 | 46.0 | 55.1 |
| 10 | Game Theory | 7.6 | 92.0 | 92.3 | 91.9 | 91.2 | 90.9 |
| 11 | HCI | 43.0 | 89.1 | 86.5 | 91.6 | 87.9 | 88.1 |
| 12 | Information Theory | 79.3 | 77.2 | 35.1 | 83.6 | 73.0 | 76.0 |
| 13 | Medical Informatics | 26.9 | 86.6 | 85.0 | 89.2 | 85.3 | 85.6 |
| 14 | Robotics | 91.4 | 86.7 | 55.8 | 93.4 | 79.2 | 87.5 |
| 15 | Theoretical CS | 57.0 | 84.2 | 75.5 | 89.6 | 84.3 | 84.1 |
| | AVERAGE | 51.5 | 69.7 | 64.0 | 79.6 | 70.2 | 75.5 |

Table 8: F1 scores for the Coauthor Physics dataset with 25 positive and 25 negative nodes.

| | Cluster | CLF | FD | WFD | LFD | PR | LPR |
|---|---|---|---|---|---|---|---|
| 1 | Particles, fields, gravitation, and cosmology | 87.5 | 87.2 | 74.2 | 91.9 | 87.7 | 89.6 |
| 2 | Atomic, molecular, and optical physics and quantum information | 69.2 | 69.5 | 63.7 | 82.7 | 73.8 | 77.9 |
| 3 | Condensed matter and materials physics | 90.5 | 81.1 | 88.1 | 95. | 89.9 | 93.3 |
| 4 | Nuclear physics | 55.2 | 81.6 | 79.4 | 87.3 | 82.6 | 84.6 |
| 5 | Statistical, nonlinear, biological, and soft matter physics | 60.4 | 46.9 | 48.3 | 74. | 58.8 | 72.6 |
| | AVERAGE | 72.6 | 73.2 | 70.7 | 86.2 | 78.6 | 83.6 |

Table 9: F1 scores for the Amazon Photo dataset with 25 positive and 25 negative nodes.

| | Cluster | CLF | FD | WFD | LFD | PR | LPR |
|---|---|---|---|---|---|---|---|
| 1 | Film Photography | 37.2 | 89.9 | 82.6 | 90.0 | 87.8 | 89.1 |
| 2 | Digital Cameras | 69.5 | 81.6 | 76.9 | 82.0 | 79.5 | 79.8 |
| 3 | Binoculars | 59.2 | 97.4 | 96.7 | 96.9 | 96.6 | 97.1 |
| 4 | Lenses | 62.7 | 64.9 | 66.2 | 73.0 | 64.9 | 70.4 |
| 5 | Tripods & Monopods | 65.7 | 82.8 | 83.1 | 90.4 | 78.2 | 84.1 |
| 6 | Video Surveillance | 71.4 | 98.3 | 98.1 | 98.7 | 98.3 | 98.3 |
| 7 | Lighting & Studio | 66.4 | 47.3 | 62.6 | 46.6 | 80.4 | 82.9 |
| 8 | Flashes | 30.9 | 56.5 | 60.2 | 67.8 | 48.2 | 55.1 |
| | AVERAGE | 57.9 | 77.3 | 78.3 | 80.7 | 79.2 | 82.1 |

Table 10: F1 scores for the Amazon Computers dataset with 25 positive and 25 negative nodes.

| | Cluster | CLF | FD | WFD | LFD | PR | LPR |
|---|---|---|---|---|---|---|---|
| 1 | Desktops | 23.5 | 60.5 | 70.8 | 68.6 | 72.2 | 80.3 |
| 2 | Data Storage | 52.2 | 39.0 | 41.1 | 44.2 | 54.7 | 58.7 |
| 3 | Laptops | 62.3 | 93.1 | 87.6 | 91.9 | 89.1 | 88.6 |
| 4 | Monitors | 36.3 | 61.1 | 64.5 | 81.1 | 59.7 | 74.0 |
| 5 | Computer Components | 72.7 | 79.9 | 75.2 | 79.7 | 76.0 | 79.2 |
| 6 | Video Projectors | 45.3 | 95.2 | 95.2 | 95.0 | 94.5 | 94.3 |
| 7 | Routers | 27.9 | 59.3 | 53.4 | 60.9 | 58.0 | 59.4 |
| 8 | Tablets | 43.4 | 89.8 | 85.7 | 89.1 | 87.9 | 86.6 |
| 9 | Networking Products | 57.2 | 64.3 | 55.5 | 70.1 | 61.6 | 65.4 |
| 10 | Webcams | 25.8 | 89.6 | 83.4 | 89.7 | 86.7 | 86.8 |
| | AVERAGE | 44.7 | 73.2 | 71.2 | 77.0 | 74.1 | 77.3 |

## D.4 DETAILED RESULTS OF EXPERIMENT WITH SAMPLING HEURISTIC

This subsection outlines the second experiment conducted with real-world data. In this experiment, a single seed node is provided without any access to ground-truth data or a pre-trained classifier. As outlined in section 4.2, our adopted heuristic approach begins with executing an initial flow diffusion process from the provided seed node. In all reported single-seed diffusion processes, we increase the amount of mass used from twice to ten times the volume of the target cluster. The 100 nodes with the highest and lowest flow diffusion embeddings are designated as positive and negative nodes, respectively. This data is used to train a classifier, as described in the previous experimental setting, and diffusion is then run from the positive nodes, followed by a sweep-cut procedure. The following tables report the results for each dataset, broken-down by their clusters.

Table 11: F1 scores for the Cora dataset with 25 positive and 25 negative nodes.

|   | Cluster | CLF | FD | WFD | LFD | PR | LPR |
|---|---------|-----|-----|-----|-----|-----|-----|
| 1 | Case Based | 44.1 | 70.3 | 70.7 | 70.4 | 71.0 | 69.2 |
| 2 | Genetic Algorithms | 60.0 | 91.8 | 91.8 | 92.6 | 90.5 | 90.5 |
| 3 | Neural Networks | 57.1 | 69.4 | 69.6 | 68.2 | 67.9 | 67.5 |
| 4 | Probabilistic Methods | 48.6 | 66.0 | 66.5 | 68.8 | 72.7 | 72.5 |
| 5 | Reinforcement Learning | 44.5 | 77.4 | 77.2 | 77.2 | 76.4 | 75.0 |
| 6 | Rule Learning | 32.4 | 71.5 | 71.5 | 71.1 | 69.8 | 69.4 |
| 7 | Theory | 41.5 | 60.8 | 60.5 | 60.0 | 60.3 | 60.4 |
|   | AVERAGE | 46.9 | 72.5 | 72.5 | 72.6 | 72.7 | 72.1 |

Table 12: F1 scores for the Pubmed dataset with 25 positive and 25 negative nodes.

|   | Cluster | CLF | FD | WFD | LFD | PR | LPR |
|---|---------|-----|-----|-----|-----|-----|-----|
| 1 | Diabetes Mellitus (Experimental) | 75.9 | 49.3 | 49.4 | 61.7 | 53.2 | 58.9 |
| 2 | Diabetes Mellitus Type 1 | 70.1 | 77.8 | 77.9 | 77.2 | 74.5 | 75.5 |
| 3 | Diabetes Mellitus Type 2 | 65.2 | 66.1 | 66.1 | 66.8 | 64.1 | 65.7 |
|   | AVERAGE | 70.4 | 64.4 | 64.5 | 68.6 | 63.9 | 66.7 |

Table 13: F1 scores for the Coauthor CS dataset in the absence of ground-truth data

|   | Cluster | FD (single-seed) | WFD (single-seed) | FD (multi-seed) | WFD (multi-seed) | LFD | PR (single-seed) | PR (multi-seed) | LPR |
|---|---------|------|------|------|------|-----|------|------|-----|
| 1 | Bioinformatics | 34.1 | 35.5 | 23.7 | 28.3 | 34.1 | 31.6 | 26.5 | 45.2 |
| 2 | Machine Learning | 29.0 | 22.5 | 21.5 | 26.9 | 25.6 | 30.8 | 28.9 | 44.3 |
| 3 | Computer Vision | 34.5 | 18.5 | 39.2 | 28.0 | 63.5 | 48.0 | 49.2 | 59.0 |
| 4 | NLP | 53.5 | 58.1 | 37.8 | 47.8 | 54.5 | 46.3 | 43.0 | 66.5 |
| 5 | Graphics | 28.6 | 43.2 | 60.5 | 61.5 | 72.0 | 58.2 | 63.1 | 69.8 |
| 6 | Networks | 42.1 | 32.5 | 46.3 | 48.2 | 71.6 | 50.7 | 54.5 | 62.3 |
| 7 | Security | 31.9 | 34.0 | 27.8 | 31.0 | 34.3 | 30.7 | 31.1 | 53.4 |
| 8 | Databases | 27.5 | 23.1 | 56.8 | 61.0 | 73.5 | 60.0 | 67.3 | 74.3 |
| 9 | Data mining | 26.6 | 17.4 | 12.9 | 19.5 | 21.4 | 30.0 | 28.1 | 40.7 |
| 10 | Game Theory | 83.1 | 82.0 | 83.0 | 81.0 | 83.0 | 58.8 | 84.4 | 85.6 |
| 11 | HCI | 68.7 | 83.6 | 79.9 | 81.0 | 87.6 | 75.6 | 82.0 | 86.4 |
| 12 | Information Theory | 36.3 | 12.3 | 56.2 | 26.0 | 78.0 | 61.5 | 65.9 | 70.3 |
| 13 | Medical Informatics | 80.9 | 76.4 | 73.9 | 72.2 | 83.4 | 75.3 | 78.1 | 83.6 |
| 14 | Robotics | 36.9 | 32.0 | 68.0 | 27.1 | 82.9 | 64.8 | 67.7 | 73.0 |
| 15 | Theoretical CS | 43.9 | 28.1 | 70.6 | 67.0 | 81.3 | 69.5 | 76.6 | 78.7 |
|   | AVERAGE | 43.8 | 39.9 | 50.5 | 47.1 | 63.1 | 52.8 | 56.4 | 66.2 |

Table 14: F1 scores for the Coauthor Physics dataset in the absence of ground-truth data

| | Cluster | FD (single-seed) | WFD (single-seed) | FD (multi-seed) | WFD (multi-seed) | LFD | PR (single-seed) | PR (multi-seed) | LPR |
|---|---|---|---|---|---|---|---|---|---|
| 1 | Particles, fields, gravitation, and cosmology | 78.5 | 60.8 | 65.5 | 48.8 | 80.9 | 72.9 | 72.5 | 80.7 |
| 2 | Atomic, molecular, and optical physics and quantum information | 38.2 | 35.1 | 45.8 | 40.9 | 58.9 | 48.9 | 50.5 | 57.5 |
| 3 | Condensed matter and materials physics | 81.5 | 84.0 | 81.8 | 81.6 | 87.4 | 84.9 | 85.4 | 87.6 |
| 4 | Nuclear physics | 63.5 | 68.6 | 60.6 | 58.9 | 77.7 | 68.7 | 68.0 | 72.1 |
| 5 | Statistical, nonlinear, biological, and soft matter physics | 52.3 | 36.4 | 23.5 | 25.6 | 59.4 | 41.8 | 33.2 | 62.5 |
| | AVERAGE | 62.8 | 57.0 | 55.5 | 51.1 | 72.9 | 63.4 | 61.9 | 72.1 |

Table 15: F1 scores for the Amazon Photo dataset in the absence of ground-truth data

| | Cluster | FD (single-seed) | WFD (single-seed) | FD (multi-seed) | WFD (multi-seed) | LFD | PR (single-seed) | PR (multi-seed) | LPR |
|---|---|---|---|---|---|---|---|---|---|
| 1 | Film Photography | 69.4 | 69.3 | 80.5 | 67.1 | 82.3 | 77.1 | 78.5 | 82.6 |
| 2 | Digital Cameras | 43.8 | 61.4 | 64.1 | 59.7 | 67.2 | 69.3 | 69.2 | 69.8 |
| 3 | Binoculars | 97.2 | 94.6 | 87.0 | 81.7 | 80.4 | 86.8 | 83.0 | 86.2 |
| 4 | Lenses | 33.5 | 35.2 | 36.0 | 37.7 | 45.5 | 41.3 | 42.2 | 49.7 |
| 5 | Tripods & Monopods | 34.3 | 36.6 | 53.6 | 69.8 | 71.9 | 50.0 | 54.0 | 61.9 |
| 6 | Video Surveillance | 98.3 | 98.1 | 98.3 | 97.9 | 98.8 | 98.1 | 98.2 | 97.0 |
| 7 | Lighting & Studio | 39.3 | 46.7 | 46.8 | 54.1 | 50.7 | 58.7 | 56.9 | 59.1 |
| 8 | Flashes | 19.8 | 17.2 | 30.3 | 33.1 | 38.1 | 32.1 | 28.8 | 34.4 |
| | AVERAGE | 54.5 | 57.4 | 62.1 | 62.6 | 66.8 | 64.2 | 63.8 | 67.6 |

Table 16: F1 scores for the Amazon Computers dataset in the absence of ground-truth data

| | Cluster | FD (single-seed) | WFD (single-seed) | FD (multi-seed) | WFD (multi-seed) | LFD | PR (single-seed) | PR (multi-seed) | LPR |
|---|---|---|---|---|---|---|---|---|---|
| 1 | Desktops | 43.1 | 47.6 | 40.7 | 41.7 | 44.3 | 41.6 | 43.1 | 44.3 |
| 2 | Data Storage | 28.8 | 32.1 | 30.6 | 30.1 | 32.2 | 38.0 | 35.6 | 40.0 |
| 3 | Laptops | 77.6 | 69.6 | 73.5 | 69.8 | 80.3 | 71.1 | 74.8 | 78.7 |
| 4 | Monitors | 32.8 | 32.6 | 38.7 | 41.0 | 53.8 | 31.0 | 37.1 | 50.2 |
| 5 | Computer Components | 54.1 | 57.3 | 73.4 | 58.7 | 73.1 | 68.1 | 68.5 | 68.8 |
| 6 | Video Projectors | 95.1 | 94.2 | 95.1 | 94.9 | 94.9 | 92.6 | 94.3 | 94.4 |
| 7 | Routers | 40.5 | 33.1 | 36.9 | 33.4 | 34.4 | 42.4 | 46.6 | 47.6 |
| 8 | Tablets | 79.0 | 67.4 | 72.9 | 68.8 | 71.6 | 69.5 | 73.3 | 73.7 |
| 9 | Networking Products | 27.4 | 29.8 | 37.9 | 31.1 | 36.7 | 46.2 | 48.2 | 50.2 |
| 10 | Webcams | 83.4 | 69.1 | 82.0 | 76.0 | 82.6 | 76.7 | 82.2 | 84.0 |
| | AVERAGE | 56.2 | 53.3 | 58.2 | 54.6 | 60.4 | 57.7 | 60.4 | 63.2 |

Table 17: F1 scores for the Cora dataset in the absence of ground-truth data

| Cluster | FD (single-seed) | WFD (single-seed) | FD (multi-seed) | WFD (multi-seed) | LFD | PR (single-seed) | PR (multi-seed) | LPR |
|---|---|---|---|---|---|---|---|---|
| 1 Case Based | 36.0 | 36.4 | 55.1 | 55.4 | 58.4 | 49.0 | 53.4 | 60.9 |
| 2 Genetic Algorithms | 30.8 | 31.6 | 88.7 | 88.9 | 90.5 | 82.8 | 88.1 | 89.5 |
| 3 Neural Networks | 45.9 | 46.3 | 43.8 | 43.7 | 40.3 | 56.4 | 59.0 | 55.9 |
| 4 Probabilistic Methods | 33.7 | 34.0 | 37.6 | 38.0 | 38.2 | 41.9 | 39.4 | 41.6 |
| 5 Reinforcement Learning | 25.3 | 26.2 | 67.7 | 67.7 | 68.0 | 64.4 | 69.2 | 69.4 |
| 6 Rule Learning | 34.4 | 33.9 | 52.4 | 52.3 | 57.3 | 48.5 | 55.0 | 59.6 |
| 7 Theory | 27.2 | 27.2 | 42.3 | 42.2 | 42.8 | 46.7 | 45.7 | 47.4 |
| AVERAGE | 33.3 | 33.7 | 55.4 | 55.4 | 56.5 | 55.7 | 58.5 | 60.6 |

Table 18: F1 scores for the Pubmed dataset in the absence of ground-truth data

| Cluster | FD (single-seed) | WFD (single-seed) | FD (multi-seed) | WFD (multi-seed) | LFD | PR (single-seed) | PR (multi-seed) | LPR |
|---|---|---|---|---|---|---|---|---|
| 1 Diabetes Mellitus (Experimental) | 34.8 | 35.0 | 37.0 | 37.0 | 43.9 | 43.9 | 42.1 | 49.8 |
| 2 Diabetes Mellitus Type 1 | 69.7 | 69.8 | 71.6 | 71.6 | 69.7 | 70.2 | 69.5 | 71.3 |
| 3 Diabetes Mellitus Type 2 | 54.6 | 55.0 | 53.1 | 53.1 | 52.4 | 54.1 | 53.1 | 55.2 |
| AVERAGE | 53.0 | 53.2 | 53.9 | 53.9 | 55.3 | 56.1 | 54.9 | 58.8 |

