# OpenReview forum: "Local Graph Clustering with Noisy Labels"
_ICLR.cc/2024/Conference — ICLR 2024 poster_

### Official Review · Reviewer_PWgh · 2023-10-28

**Soundness:** 3 good
**Presentation:** 3 good
**Contribution:** 2 fair
**Rating:** 6
**Confidence:** 2

**Summary:**

The paper proposes a method that leverages noisy node labels for local graph clustering. In this approach, edge weights are defined based on the noisy labels in the graph, giving higher weight to nodes with the same noisy label and lower weight to nodes with different labels. Using this weighted graph, the authors apply a graph diffusion local clustering method called 'flow diffusion' to determine the clustering. The paper includes both theoretical proofs and empirical evidence, demonstrating why this approach improves clustering accuracy compared to applying the flow diffusion method directly to the original graph.

**Strengths:**

The proposed method for local graph clustering is conceptually simple and easy to implement while apparently being effective. It is also flexible, as it can be combined with different diffusion-based local clustering algorithms. In addition, the authors provide theoretical guarantees for its success under mild conditions.

**Weaknesses:**

The experimental section lacks a comparison with alternative methods that are not flow-based.

The paper lacks a detailed exploration of the algorithm's sensitivity to hyperparameters. While the robustness to the choice of $\epsilon\in[0.01,0.1]$ is briefly mentioned, there is no reference to the robustness of the source mass of the seed ($\Delta_s$). For the experiments with the synthetic datasets, different $\Delta_s$ values are tested but only the best is provided. The paper would benefit if there was an study on the choice of the hyperparameters.

The discussion regarding Theorem 3.4 is not adequately understandable. See questions for more details.

**Questions:**

- Regarding the discussions following Theorem 3.4:

  - The statement that "as $\gamma$ becomes lager, it generally becomes more
    difficult to accurately recover K [the cluster]" appears counterintuitive. Given that $\gamma\coloneqq \frac{p(k-1)}{q(n-k)}$ signifies the ratio of internal to external connections within a cluster, an increase in $\gamma$ implies more internal connections. This suggests a stronger, more cohesive cluster. Could you elaborate on why higher $\gamma$ values lead to increased difficulty in cluster recovery?

  - The assertion that "F1 is lower bounded by a constant as long as $\gamma$ is a constant" raises questions.  What does "$\gamma$ is constant" mean in this context? As far as I understand, the value $\gamma$ is defined for each node and implicitly depends exclusively on the cluster size, $k$, and the edge probabilities, $p$ and $q$. I suggest that this dependence is stress further, since as it is written now, $\gamma$ is apparently a constant by itself. Consequently, I guess that "$\gamma$ constant" means that it has the same value for all nodes (or for all clusters). Nonetheless, the lower bound of F1 would still depend on $a_0$ and $a_1$. Could you clarify why the lower bound of F1 is described as a constant despite potential dependencies on other parameters?

  - The statement, "even when the initial labels are deemed fairly accurate based on $a_0$ and $a_1$, e.g. $a_1$ = 1, $a_0$ = 0.99, the F1 score of the labels can still be very low." Could you explain the factors contributing to the low F1 score in this scenario?

- How robust is the proposed method concerning the source mass of the seed ($\Delta_s$)?

---

> ### Author Response · Authors · 2023-11-17
> **Response (Part 1 of 2)**
>
> We sincerely thank the reviewer for reading our manuscript and providing us with valuable feedback. We are glad that the reviewer likes the simplicity of our method. Designing a simple, effective and robust method that works in a novel problem setting is the major focus of this paper. We address all questions and concerns below.
>
> **1. Comparison with methods that are not flow-based**
> > The experimental section lacks a comparison with alternative methods that are not flow-based.
>
> In Appendix D.2 (previously Appendix B.2) we compared with PageRank, which is not a flow-based method. Our results demonstrate that the label-reweighting scheme (Equation 4) effectively helps PageRank improve local clustering accuracy. We would like to mention that we are the first to consider the problem setting of local clustering with noisy labels. Apart from global methods which require the costly operation of processing the entire graph, there is no other local method that works in this setting. Nonetheless, to further demonstrate the effectiveness of our local method, we added a comparison with Graph Convolutional Networks (GCNs) to Appendix C.3 in the updated manuscript. GCNs are a strong baseline for semi-supervised classification. However, GCNs are not scalable as their runtime increases with the size of the graph. The new results in Appendix C.3 demonstrate that our method is much faster while being much more accurate.
>
> **2. Hyperparameter analysis**
> > The paper lacks a detailed exploration of the algorithm's sensitivity to hyperparameters.
>
> We updated the manuscript by adding a study on hyperparameter choices from both a theoretical perspective and a practical perspective. The theoretical discussion is provided in Appendix B and the empirical study is provided in Appendix C. On the theoretical side, we added discussions on how to set $\epsilon$ when the graph is generated according to the local random model in Definition 3.2. On the empirical side, we tested the robustness of our method against various choices for both $\epsilon$ and $\theta$. We invite the reviewer to take a look at our new results in Appendix C of the updated manuscript, which verify our claim that the clustering accuracy is not very sensitive to hyperparameters.
>
> Let us discuss more on the choice of $\theta$, i.e. the amount of source mass at the seed node. Roughly speaking, the parameter $\theta$ controls the size of the subgraph around the seed node that diffusion will explore. It is therefore closely related to the size of the target cluster. In practice, one just needs to have a very rough estimate of the size of the target cluster, and then set $\theta$ to be a multiple (say 2,3,4,5, etc) of the target size. Our robustness results in Appendix C.1 indicate that the clustering accuracy does not change much as long as $\theta$ is set within a reasonable range.
>
> **3. Theorem 3.4**
> > The statement that "as $\gamma$ becomes larger, it generally becomes more difficult to accurately recover K [the cluster]" appears counterintuitive. Given that $\gamma := \frac{p(k-1)}{q(n-k)}$ signifies the ratio of internal to external connections within a cluster, an increase in $\gamma$ implies more internal connections. This suggests a stronger, more cohesive cluster. Could you elaborate on why higher $\gamma$ values lead to increased difficulty in cluster recovery?
>
> That was indeed a typo in our original manuscript. The reviewer is right. As $\gamma$ increases, we have a stronger internal signal, and hence it becomes easier to recover the target cluster K. We have fixed the typo in the updated manuscript.

---

> ### Author Response · Authors · 2023-11-17
> **Response (Part 2 of 2)**
>
> > The assertion that "F1 is lower bounded by a constant as long as $\gamma$ is a constant" raises questions. What does "$\gamma$ is constant" mean in this context? As far as I understand, the value $\gamma$ is defined for each node and implicitly depends exclusively on the cluster size, $k$, and the edge probabilities, $p$ and $q$. I suggest that this dependence is stress further, since as it is written now, $\gamma$ is apparently a constant by itself. Consequently, I guess that "$\gamma$ constant" means that it has the same value for all nodes (or for all clusters). Nonetheless, the lower bound of F1 would still depend on $a_0$ and $a_1$. Could you clarify why the lower bound of F1 is described as a constant despite potential dependencies on other parameters?
>
> In the context of local graph clustering or cluster recovery, we treat the number of nodes in the graph, i.e. $n$, as a variable. Therefore, other quantities $p, q$, and $k$ are considered functions of $n$. When we say a quantity is constant, we mean that its value is not affected by the size of the graph. For example, if $k$ is a constant (does not change with $n$), $p = \frac{\log k}{k-1}$, and $q = \frac{\log n}{n-k}$, then $\gamma = \frac{\log k}{\log n}$ which is not a constant with respect to $n$. Or, if $k = \log n$, $p = \frac{\log k}{k-1}$, and $q = \frac{1}{n-k}$, then $\gamma = \log\log n$ which is again not a constant with respect to $n$. We revised the wording by requiring that $\gamma=\Omega_n(1)$. When we say F1 is lower bounded by a constant, we mean that the lower bound in Equation 5 does not go to 0 as $n$ goes to infinity. We thank the reviewer for raising this question and hope that it clarifies the confusion.
>
> > The statement, "even when the initial labels are deemed fairly accurate based on $a_0$ and $a_1$, e.g. $a_1=1, a_0=0.99$, the F1 score of the labels can still be very low." Could you explain the factors contributing to the low F1 score in this scenario?
>
> Let us assume that the size of the target cluster $k$ is a constant with respect to $n$, for example, $k = 10,000$. When the label accuracy is $a_1 = 1$ and $a_0 = 0.99$, the number of true positives is $k$, the number of false positives is $(1-a_0)(n-k)=0.01(n-k)$, and the number of false negatives is $(1-a_1)k=0$. Therefore, the F1 score is $[1 + 0.01(n-k)]^{-1}$. For sufficiently large graphs, for example, $n > 101k$, the F1 becomes smaller than $[1+k]^{-1} < 0.0001$.
>
> ---
> We thank the reviewer again and invite any further questions. We hope that, if satisfied, the reviewer will consider raising their score.

---

> ### Author Response · Authors · 2023-11-21
> **Is there anything else we can address?**
>
> Would the reviewer kindly confirm that our response has adequately addressed all questions, or else kindly let us know what remains unclear? If the reviewer is satisfied, we would appreciate if they could consider raising the score.

---

> > ### Comment · Reviewer_PWgh · 2023-11-22
> >
> > I thank the authors for their insights. I keep my score.

---

> > > ### Author Response · Authors · 2023-11-22
> > >
> > > We thank the reviewer for their time and thoughtful feedback which helped us improve the manuscript.

---

### Official Review · Reviewer_oYEs · 2023-11-01

**Soundness:** 3 good
**Presentation:** 3 good
**Contribution:** 3 good
**Rating:** 6
**Confidence:** 3

**Summary:**

In this paper, the authors study the local graph clustering problem. They investigate the benefits of incorporating noisy labels for local graph clustering. By constructing a weighted graph with such labels, they study the performance of graph diffusion-based local clustering method on both the original and the weighted graphs. The experimental results demonstrate the effectiveness of the proposed methods.

**Strengths:**

S1. The problem studied in the paper is important.
S2. The performance of flow diffusion over a random graph model is analyzed.
S3. Empirical experiments are conducted to evaluate the proposed methods.

**Weaknesses:**

W1. It is unclear whether the proposed method needs to explore the entire graph.
W2. The relation between the additional node information like texts, images and the proposed methods is unclear.
W3. Whether the compared methods in the experiment section are comprehensive or not is unclear.

**Questions:**

Q1. In Section 1, the authors claim that the task of local graph clustering aims to
identify a small cluster of nodes that contains all or most of the seed nodes, without exploring the entire graph. However, it is unclear whether the proposed method needs to explore the entire graph.

Q2. In Section 1, the authors claim that the additional information like texts, images can significantly benefit clustering, but the relation between the additional information and the proposed methods is unclear.

Q3. Only flow diffusion and Label-based PageRank methods are compared in the experiments, it is unclear whether the compared methods in the experiment section are comprehensive or not.

Q4. It is unclear whether the assumption on a_0 and a_1 is practical in real applications.

---

> ### Author Response · Authors · 2023-11-17
> **Response**
>
> We sincerely thank the reviewer for their reading of the manuscript and the feedback. We address all questions and concerns below.
>
> **1. Locality of the algorithm**
> > Q1. In Section 1, the authors claim that the task of local graph clustering aims to identify a small cluster of nodes that contains all or most of the seed nodes, without exploring the entire graph. However, it is unclear whether the proposed method needs to explore the entire graph.
>
> Our method does not require accessing the entire graph for the following reasons. First, flow diffusion does not require exploring the entire graph [1,2]. Second, our label-weighting scheme in Equation 4 does not need to be applied to every edge in the graph, instead, in a localized implementation, the label-weighting scheme is only applied to an edge whenever the edge is used by flow diffusion. Due to the locality nature of flow diffusion [1,2], the number of edges we need to weigh is independent of the size of the graph. We updated Remark 3.3 in the manuscript to clarify this point. In the empirical setting where we need to train a classifier and obtain noisy labels from its predictions, the complexity of obtaining such a classifier from a fixed number of training samples is constant with respect to the size of the graph. Again, this procedure does not require access to the entire graph.
>
> In the updated manuscript, we added an empirical runtime analysis to Appendix C.2. Our method has an average running time 0.13 seconds. The fast running time of our method is a direct result of our method being a local method, i.e., it only requires using a small part of the graph around the seed node.
>
> **2. Exploiting additional information**
> > Q2. In Section 1, the authors claim that the additional information like texts, images can significantly benefit clustering, but the relation between the additional information and the proposed methods is unclear.
>
> As discussed in the abstract and introduction, additional node information does not necessarily come as node labels. In practice, additional node information are often high-dimensional vector representations of texts or images that describe the properties of a node. In this case, we can obtain a mapping that converts these high-dimensional vector representations into noisy node labels. For example, in our experiments, the real-world data contain node attributes which are vector representations of the text description of a node. In addition, we have a few ground-truth labels that reveal cluster affiliation for some nodes. This allows us to build a classifier and use its predictions as noisy node labels. Let us emphasize that the entire process is still local and does not require processing the entire graph.
>
> **3. Comprehensiveness in empirical evaluation**
> > Q3. Only flow diffusion and Label-based PageRank methods are compared in the experiments, it is unclear whether the compared methods in the experiment section are comprehensive or not.
>
> The goal of our empirical evaluation is to demonstrate the advantage of employing diffusion over the label-weighted graph. In other words, we would like to empirically verify that the label-weighting scheme (Equation 4) can help local clustering. We selected two representative diffusion methods. Flow diffusion is selected for its good empirical performance demonstrated in prior work [1,2], and PageRank is selected for its popularity. For each of these methods, our empirical evaluation covers six popular benchmark datasets, which in total contain 58 unique clusters. The volume of the experiments and the consistency in empirical results adequately demonstrate the effectiveness of our method.
>
> To further demonstrate the effectiveness of our method against global methods (which require processing the entire graph), we added a comparison with Graph Convolutional Networks in Appendix C.3 in the updated manuscript. The new results again highlight the fast runtime and superior accuracy of our local approach.
>
> **4. Assumption on $a_0$ and $a_1$**
> > Q4. It is unclear whether the assumption on $a_0$ and $a_1$ is practical in real applications.
>
> Our assumption simply requires that the label accuracy is better than random guess. Therefore, it is a reasonable assumption in practice. To empirically verify this point with real data, we randomly selected one of our empirical settings over real-world data (Coauthor CS dataset with 20 supervised points). The average label accuracies across 1500 trials are $a_0=75.2$ and $a_1= 90.4$ which are well above 1/2.
>
> [1] p-Norm Flow Diffusion for Local Graph Clustering. K. Fountoulakis, D. Wang, S. Yang. ICML 2020.\
> [2] Weighted Flow Diffusion for Local Graph Clustering with Node Attributes: an Algorithm and Statistical Guarantees. S. Yang, K. Fountoulakis. ICML 2023
>
> ---
> We thank the reviewer again and invite any further questions. We hope that, if satisfied, the reviewer will consider raising their score.

---

> > ### Comment · Reviewer_oYEs · 2023-11-23
> >
> > I thank the authors for their response. I keep my score.

---

> ### Author Response · Authors · 2023-11-21
> **Is there anything else we can address?**
>
> Would the reviewer kindly confirm that our response has adequately addressed all questions, or else kindly let us know what remains unclear? If the reviewer is satisfied, we would appreciate if they could consider raising the score.

---

### Official Review · Reviewer_CYJN · 2023-11-04

**Soundness:** 3 good
**Presentation:** 4 excellent
**Contribution:** 3 good
**Rating:** 8
**Confidence:** 3

**Summary:**

This paper studies local graph clustering (identifying a small cluster containing the seed nodes) with noisy node labels. The authors propose a localized algorithm based on flow diffusion on a weighted graph constructed from the noisy labels. For a particular local random model, it is shown that the proposed algorithm yields an accurate recovery of the target cluster with high probability measured by F1 score. The authors also demonstrate through synthetic data as well as several real-world datasets that the approach has better performance compared to using edges or labels alone.

**Strengths:**

1. The problem of recovering a small cluster with additional noisy labels is new and interesting.
2. The proposed method, though based on a simple modification of flow diffusion, is localized and computationally efficient.
3. The theoretical analysis is sound and convincing. Moreover, it addresses the situation when the F1 score of labels is very low.
4. The experiments further justify the findings and show significant improvements over using only edges or labels.

**Weaknesses:**

1. Flow diffusion has been studied extensively under various settings in the literature. The current work is a direct application of it.
2. The theoretical analysis is limited to a very simple random graph model in specific parameter regimes, and there is room for stronger arguments (see question 2-5).
3. The choice of several parameters in the algorithm probably requires more investigation and discussion (see question 1).

**Questions:**

1.  As noted by the authors, $\epsilon$ interpolates between two special scenarios that are suitable for different level of label noise. From a theoretical perspective, how should $\epsilon$ be chosen if $a_0$ and $a_1$ are known? Also, is there possibly a way to estimate $\epsilon$ and $\theta^\dagger$ from the graph?
2. The main theorem requires $p = \omega(\frac{\sqrt{\log k}}{\sqrt{k}})$ which produces a dense cluster. Can this condition be relaxed to include less dense clusters?
3. Is there an information-theoretic limit on recovering local graph cluster with noisy labels such that the sharpness of the lower bound on F1 can be evaluated?
4. It would be helpful to compare the current result with these on local graph clustering without labels. In particular, when the labels do not provided any information ($a_0 = a_1 = 1/2$), does the proposed algorithm have improvements over the previous ones?
5. Weighted message passing is another localized algorithm applied to SBM with noisy label information (see [1] and references therein). How does the flow diffusion algorithm and theoretical result compare to it?

[1] Cai, T. T., Liang, T., & Rakhlin, A. (2020). Weighted message passing and minimum energy flow for heterogeneous stochastic block models with side information. The Journal of Machine Learning Research, 21(1), 346-379.

---

> ### Author Response · Authors · 2023-11-17
> **Response (Part 1 of 2)**
>
> We sincerely thank the reviewer for their reading of the manuscript and detailed feedback. We are glad that they note the novelty of our problem setting and the effectiveness of our method. We answer their interesting questions below.
>
> **1. Parameter choice**
> > As noted by the authors, $\epsilon$ interpolates between two special scenarios that are suitable for different level of label noise. From a theoretical perspective, how should $\epsilon$ be chosen if $a_0$ and $a_1$ are known? Also, is there possibly a way to estimate $\epsilon$ and $\theta^\dagger$ from the graph?
>
> This is an excellent question. It turns out that an “optimal” $\epsilon$ will depend on model parameters $n, k, p, q$, label accuracy $a_0,a_1$, and the clustering metric one aims to maximize. For example, if the goal is to maximize recall, then one should simply set $\epsilon=1$, that is, work with the original graph. However, if the goal is to obtain a recovery that balances precision and recall, for example, maximize the F1 score, then it becomes nontrivial to characterize a good choice for $\epsilon$. We updated the manuscript by adding a separate Appendix B to discuss the conditions under which one should prefer setting $\epsilon=0$ or $\epsilon>0$. In summary, we provided two conjectures:
>
> **C1:** If $(1-a_1)pk > a_0q(n-k)$, then one should set $\epsilon > 0$.\
> **C2:** If $(1-a_1)p^2k < a_0q^2(n-k)$, then one should set $\epsilon = 0$.
>
> We provided an informal analysis of these conditions based on the average behavior of flow diffusion. In addition, we empirically verify the conditions over synthetic data (see the new Appendix B.1 in the updated manuscript). We note that there is a gap of order p/q between the two conditions. Closing the gap and characterizing an actual value for epsilon is an interesting question that requires more careful analysis. We leave it for future work.
>
> As for the choice of $\theta$, this is a common parameter for diffusion-based local clustering algorithms. For example, $\theta$ in our method plays a similar role as the tolerance parameter in Approximate Personalized PageRank. All that we require is that the user has a rough estimate of the size of the target cluster they wish to recover, and then set $\theta$ to be a multiple (say 2,3,4,5, etc.) of the target size. On the practical side, the flexibility to choose $\theta$ allows the user to control the size of the output cluster. This is advantageous when they already have a rough idea on the number of nodes they want to recover.
>
> Finally, from an empirical perspective, we invite the reviewer to take a look at the new Appendix C.1 in the updated manuscript. We tested the robustness of our method against different choices of $\epsilon$ and $\theta$ over real-world data. It turns out that the results are not very sensitive to parameter choices, as long as they are within a reasonable range.
>
> **2. Sparser clusters**
> > The main theorem requires $p = \omega(\frac{\sqrt{\log k}}{\sqrt{k}})$ which produces a dense cluster. Can this condition be relaxed to include less dense clusters?
>
> We can relax the assumption on $p$ if we impose a stronger assumption on $q$ and $a_1$. This is because our analysis requires the internal signal to be sufficiently stronger than the external noise. Therefore, if we have a sparser cluster, then we require fewer external connections which are controlled by $q$ and $a_1$. On the other hand, it might be possible to relax the assumption on $p$ without making additional assumptions, but that will likely require adopting a different local clustering method or a very different approach to analyze the diffusion mechanism.
>
> **3. Information-theoretic limit**
> > Is there an information-theoretic limit on recovering local graph cluster with noisy labels such that the sharpness of the lower bound on F1 can be evaluated?
>
> To the best of our knowledge, we are the first to propose a study of the local graph clustering problem with noisy labels. No information-theoretic limit has been established for this problem setting. It would be a fantastic future research to determine it. We believe that deriving such a limit would likely require additional assumptions on the connectivity of the rest of the graph. In our current theoretical setting, we did not impose any assumption on the connectivity outside the target cluster. Consequently, the lower bound on F1 we presented in Theorem 3.4 is not sharp in general. One could expect a better lower bound with additional assumptions on the edge connectivity outside the target cluster. For example, the diffusion dynamics will be different when $V\backslash K$ is a clique versus when $V\backslash K$ induces no edges.

---

> ### Author Response · Authors · 2023-11-17
> **Response (Part 2 of 2)**
>
> **4. Local clustering without labels**
> > It would be helpful to compare the current result with those on local graph clustering without labels. In particular, when the labels do not provided any information ($a_0=a_1=1/2$), does the proposed algorithm have improvements over the previous ones?
>
> This is a great question. If we compare apples to apples, that is, if we compare flow diffusion with and without labels, Theorem 3.4 provides a sufficient condition on $a_0$ and $a_1$ such that diffusion with labels is better. The condition is a bit strong because we did not have any assumption on the connectivity outside the target cluster. To make the comparison more clear, and as the reviewer suggested, let us take a look at the case when $a_0 = a_1 = 1/2$. We may apply Theorem 3.5 from [2] and get that flow diffusion without labels has an F1 lower bounded by $[1+\frac{1+2\gamma}{2\gamma^2}]^{-1}$, whereas Theorem 3.4 in our manuscript says that flow diffusion with labels has an F1 lower bounded by $[1+\frac{(1+\gamma)^2)}{4\gamma^2}]^{-1}$. When $\gamma < 3.89$, using labels gives a better lower bound. Note that a small gamma means that the internal signal is not much stronger than the external noise, in this case, it makes sense that reducing both internal and external connection by $1/2$ might give a better result. However, let us mention that the lower bounds may not be tight, and it’s not a good idea to compare lower bound with lower bound. Equation 6 in our Theorem 3.4 was obtained by comparing the lower bound using labels versus the upper bound without using labels.
>
> **5. Comparison with Weighted Message Passing**
> > Weighted message passing is another localized algorithm applied to SBM with noisy label information (see [1] and references therein). How does the flow diffusion algorithm and theoretical result compare to it?
>
> We thank the reviewer for suggesting the excellent reference. Let us start by explaining the two notions of locality. In [1] and the references therein, an algorithm is called local if the computation at any node in the graph does not require querying a node far away from that node. For example, belief propagation is considered a local algorithm with such a definition of locality. The motivation for this definition of locality stems from distributed computing. In our paper and related literature on local graph clustering, an algorithm is called local if the total computation time does not depend on the size of the graph. Under this definition of locality, belief propagation is not a local algorithm, since it requires computation at every node in the graph. Such a definition of locality stems from the need to probe internet-scale networks with limited space and time. Weighted Message Passing (Algorithm 1 in [1]) requires constructing a BFS tree at every node in the graph, this immediately takes $\Omega(n^2)$ time, which makes it a non-local method according to our definition of locality. Weighted Message Passing uses more information (i.e. it carries out computation at every node in the graph, and in [1] the analysis uses assumption about edge connectivity outside a fixed target cluster, which we do not have in our setting), and therefore [1] presents a more complete set of theoretical results. On the other hand, our method is much simpler, easier to implement, and has a faster running time both in theory and in practice.
>
> [1] Weighted message passing and minimum energy flow for heterogeneous stochastic block models with side information. T. T. Cai, T. Liang, A. Rakhlin. JMLR 2020\
> [2] Weighted Flow Diffusion for Local Graph Clustering with Node Attributes: an Algorithm and Statistical Guarantees. S. Yang, K. Fountoulakis. ICML 2023
>
> ---
> We thank the reviewer again for the careful reading of our manuscript and their insightful questions. We welcome any further questions.

---

> > ### Comment · Reviewer_CYJN · 2023-11-22
> >
> > I thank the authors for carefully addressing my comments and adding new discussions. All my questions have been properly answered. I keep my score.

---

> > > ### Author Response · Authors · 2023-11-22
> > >
> > > We thank the reviewer for their time and thoughtful feedback which helped us improve the manuscript.

---

### Official Review · Reviewer_LuBd · 2023-11-12

**Soundness:** 2 fair
**Presentation:** 2 fair
**Contribution:** 2 fair
**Rating:** 3
**Confidence:** 4

**Summary:**

This paper studies the local graph clustering task using noisy node labels. The theoretical justification is provided to investigate the benefits of incorporating noisy labels for local graph clustering. Some empirical results are shown to illustrate the method.

**Strengths:**

- Graph clustering is a very fundamental problem for graph-related problems, and exploring noisy labels perspective is a very interesting topic.
- The paper is well-organized and easy to be understood.

**Weaknesses:**

- About my confusion with the setting of the paper. The setting is local graph clustering or local graph clustering with noisy node labels? If it is local graph clustering, I think this paper tackles the local graph clustering problem with the help of noisy labels. The authors say that they abstract all available sources of additional information as noisy node labels. However, the setting of their experiments does not reflect this point, i.e., they only use the predicted labels as the noisy labels. I did not see any other available sources to transform into noisy labels. If the setting is local graph clustering with noisy node labels, the authors should compare the proposed method with other existing methods that focus on this setting, i.e., node labels should be contained in the compared methods. So the experiments have limitations. I hope the authors clarify the setting and make more comparisons with existing methods. Experiments should be enriched to verify the effectiveness of the method.
- The paper lacks complexity analysis.
- The comparison of the empirical running time of each method should be contained. Hyperparameter analysis is also missed.
- The analysis and discussion of the empirical results are clearly inadequate.

**Questions:**

See above.

---

> ### Author Response · Authors · 2023-11-17
> **Response (Part 1 of 3)**
>
> We would like to sincerely thank the reviewer for reading the manuscript. We address the reviewer’s questions and concerns one-by-one below. Please let us know if there are more questions. We are happy to discuss.
>
> ## 1. Clarification on empirical experiments and problem setting
> We start with concise answers, followed by more thorough clarifications on empirical experiments and problem setting.
>
> *Q1: Available sources to transform into noisy labels*
> >...they only use the predicted labels as the noisy labels. I did not see any other available sources to transform into noisy labels
>
> In our experiments with real-world data, we do not use the predicted labels directly. We transform available sources into noisy labels, exactly as the reviewer was anticipating. In particular, we adopt the following two-step procedure. First, we transform the additional node information, which consists of vectorized text representations, into binary predictions. Second, we use the binary predictions as noisy labels and apply our method. During the first step, we train a classifier using text representations and partial ground-truth labels. This is what we do for our first set of experiments with real-world data, and the results are shown in Figure 3. Since the number of training samples is constant, the complexity of obtaining such a classifier is constant with respect to the graph size. Consequently, the entire two-step procedure remains local.
>
> *Q2: Comparison with existing methods for local clustering with noisy labels*
> > If the setting is local graph clustering with noisy node labels, the authors should compare the proposed method with other existing methods that focus on this setting
>
> We are glad that the reviewer asked this question. Our method is the first local method that works with noisy node labels. There is no other local method that works in the problem setting. Nevertheless, we included an additional comparison to Appendix C.3 in the updated manuscript. For the reviewer’s convenience, we explain the details below.
>
> **1.1 More empirical comparisons**\
> Since we are the first to study the local graph clustering problem with noisy labels, there is no other local method to compare against under the same settings. The most relevant methods that come close to solving this problem scalably are the following. 1) Traditional local clustering methods such as Flow Diffusion (FD) and PageRank; 2) Recent local clustering methods that work with node attributes, such as Weighted Flow Diffusion (WFD); 3) Traditional, global, classification methods that work with node attributes and ground-truth node labels, such as logistic regression (shown as Classifier in Figure 3). Among these methods, 1) and 2) are local but cannot work with ground-truth node labels, 3) can work with ground-truth labels, but it requires evaluating every node in the graph. In the original version of the paper, we put significant effort to provide a comprehensive empirical comparison among these methods, covering 6 popular real-world datasets which in total include 58 unique clusters.
>
> Nevertheless, to further expand the range of comparisons beyond the traditional setting for local graph clustering, we added a new comparison with Graph Convolutional Networks (GCNs) in the updated manuscript. Given node attributes and partial ground-truth labels, GCNs are a strong baseline for semi-supervised node classification on graphs. However, they require processing the entire graph at both training and inference stages, and thus GCNs have much worse scalability than 1), 2), 3). In addition, GCNs require many ground-truth labels to work well. Under our empirical setting with limited ground-truth labels, our empirical results in Appendix C.3 indicate that our method runs much faster, while at the same time achieving a much higher F1. For the reviewer’s convenience, we summarize the results below. The reason that GCNs have poor performance here is that we only have 20 ground-truth labels and consequently GCNs tend to create balanced clusters. In the context of local graph clustering, this leads to low precision and thus lower F1. To make GCN achieve a higher F1 than LFD, we had to use 600 ground-truth labels, which is not very realistic in this context.
> ||$\qquad$LFD|$\qquad$GCN|
> |:-:|:-:|:-:|
> |F1 score|73.0|46.9|
> |Runtime|0.13 $\pm$ 0.03 s|3.68 $\pm$ 0.31 s|
>
> If the reviewer would like us to compare with additional methods, please kindly let us know. We would be happy to add more comparisons to convince the reviewer of the effectiveness of our method.

---

> ### Author Response · Authors · 2023-11-17
> **Response (Part 2 of 3)**
>
> **1.2 Problem setting**
> > The setting is local graph clustering or local graph clustering with noisy node labels? [...] I hope the authors clarify the setting
>
> We study local graph clustering with noisy node labels. Our method applies to both scenarios that the reviewer described: **Scenario 1:** The input is a graph along with given noisy node labels. **Scenario 2:** The input is a graph along with additional side information such as node attributes and partial ground-truth labels. For both scenarios, we are the first to study the local graph clustering problem under these settings. To the best of our knowledge, there is no other local method that works in these settings.
>
> Scenario 1 with given noisy node labels allows us to theoretically characterize the benefits of incorporating noisy labels for local graph clustering. Scenario 2 is a practical setting where we showcase the effectiveness of our method. After all, in practice, we often do not have direct access to noisy node labels, but instead, we have access to node attributes and partial ground-truth labels. The node attributes are often vector representations of additional information, such as texts that describe the properties of nodes. The partial ground-truth labels provide cluster affiliation information for a few nodes in the graph. This type of data is the most predominant in practice, yet no existing local clustering method can work well with such data. In our experiments, we convert side information into noisy node labels by training a classifier that predicts node labels based on node attributes. This reduces Scenario 2 to Scenario 1 and allows us to apply our method. The entire process can be implemented to maintain locality, that is, it does not require access to the entire graph. Our experiments cover six of the most frequently used benchmark datasets, and our method is consistently better.
>
> ## 2. Complexity analysis
> Given noisy node labels, our method applies flow diffusion to the label-weighted graph $G^w$. As we briefly discussed in Remark 3.3 in the manuscript, the complexity of our method is the same as the complexity of flow diffusion, whose complexity has been analyzed in previous works, see [1,2] for example. Following the discussion in [1,2], the running time complexity of our method is $O(\bar{d}\theta)$, where $\bar{d}$ is the maximum degree in the output cluster and $\theta$ is the total amount of source mass at the seed node. The running time is independent of the size of the graph. This is standard complexity analysis in the local graph clustering context. We would like to thank the reviewer for the question. We updated Remark 3.3 to make the complexity result more clear.
>
> In the empirical setting where we need to train a classifier and use its predictions as noisy node labels, the complexity depends on the number of training samples. Since we consider a fixed number of ground-truth labels, the complexity of obtaining the classifier (and hence noisy node labels) is constant with respect to the size of the graph.
>
> To empirically demonstrate the fast running time of our method, we provided additional runtime analyses in Appendix C.2. On average, the total running time of our method is 0.13 seconds. The fast runtime is due to the fact that our method is local: its running time complexity is independent of the size of the graph.
>
> [1] p-Norm Flow Diffusion for Local Graph Clustering. K. Fountoulakis, D. Wang, S. Yang. ICML 2020.\
> [2] Weighted Flow Diffusion for Local Graph Clustering with Node Attributes: an Algorithm and Statistical Guarantees. S. Yang, K. Fountoulakis. ICML 2023.

---

> ### Author Response · Authors · 2023-11-17
> **Response (Part 3 of 3)**
>
> ## 3. Runtime and hyperparameter analysis
> We added a hyperparameter analysis to Appendix C.1 in the updated manuscript, where we empirically test the robustness of our method against various choices of edge weight $\epsilon$ and the amount of source mass $\theta$. Our new results in Appendix C.1 indicate that our method is quite robust to the choice of source mass as well as the selection of the edge weight. We added a runtime analysis to Appendix C.2 in the updated manuscript. It contains a detailed runtime breakdown for each component of our method. For comparison purposes, in Appendix C.3 we also report the runtime of GCN. Our method is much faster while being much more accurate.
>
> We invite the reviewer to take a look at Appendix C in the updated manuscript for detailed results and discussions.
>
> ## 4. Empirical results
> We hope that with the new empirical comparisons and analyses, our empirical results are sufficient to demonstrate the effectiveness of our method. If the reviewer has more actionable suggestions to further improve our manuscript, please kindly let us know. We would be happy to address any further questions.
>
> ---
> We thank the reviewer again and invite them to respond with further comments. We hope the clarifications and additional empirical results will be sufficient for them to consider raising their score.

---

> ### Author Response · Authors · 2023-11-21
> **Is there any further clarification needed?**
>
> We thank the reviewer again for noting our noisy labels perspective interesting and the paper is easy to understand, albeit from a minor confusion the reviewer had with respect to our empirical experiments. Could the reviewer kindly confirm that our responses have made clear about the problem setting, and that the effectiveness of our method is verified by additional experiments? Or else kindly let us know what remains unaddressed?

---

### Meta-Review · Area_Chair_MuM9 · 2023-12-09

**Metareview:**

The paper introduces and studies the following graph clustering problem: Given a graph and a seed set of nodes, the goal is to recover an unknown cluster containing the seed nodes. To this end, we are given noisy label information for the nodes that are obtained by corrupting the ground truth clustering: we start with the 0/1 labels indicating whether the node is in the target cluster or not, and we flip a fraction of the labels to obtain the noisy labeling. The paper designs local clustering algorithms based on flow diffusion, and it provides a theoretical analysis for graphs that arise from a random graph model that can be viewed as a local version of the stochastic block model. The experimental evaluation shows that the proposed method can leverage various sources of information about the target clustering, leading to improved performance in practice.

The reviewers are generally positive about the paper, and note several main strengths of the paper:
* The problem is well-motivated and novel.
* The algorithm is conceptually simple and easy to implement and practical.
* The paper provides both theoretical and empirical support for the proposed method.

The reviewers also identified several concerns regarding the experimental evaluation:
* The set of prior methods considered in the experimental evaluation may be limited, and it is restricted to flow-based methods. The author response emphasized that this work is the first to propose a local method, and there is no other local method to compare against.
* The proposed algorithm uses several hyperparameters and it is unclear how sensitive the algorithm is to the choice of hyperparameters. The authors have added further discussion on the hyperparameters during the discussion period.

Overall, this paper makes a solid contribution to the area of graph clustering with node label information. The proposed algorithm is simple and practical and it may lead to improved performance in applications.

**Justification For Why Not Higher Score:**

The main contribution may not be of a wide enough interest to merit being highlighted as a spotlight.

**Justification For Why Not Lower Score:**

The main contribution is theoretically interesting and practically relevant, and it is a valuable addition to the area.

---

### Decision · Program_Chairs · 2024-01-16

Accept (poster)